# Improving the Scaling Laws of Synthetic Data with Deliberate Practice

**Reyhane Askari-Hemmat** [* 1]   **Mohammad Pezeshki** [* 1]   **Elvis Dohmatob** [1 2 3]   **Florian Bordes** [1]   **Pietro Astolfi** [1]
**Melissa Hall** [1]   **Jakob Verbeek** [1]   **Michal Drozdzal** [1]   **Adriana Romero-Soriano** [1 3 4 5]

## Abstract

Inspired by the principle of deliberate practice in human learning, we propose Deliberate Practice for Synthetic Data Generation (DP), a novel framework that improves sample efficiency through dynamic synthetic data generation. Prior work has shown that scaling synthetic data is inherently challenging, as naively adding new data leads to diminishing returns. To address this, pruning has been identified as a key mechanism for improving scaling, enabling models to focus on the most informative synthetic samples. Rather than generating a large dataset and pruning it afterward, DP efficiently approximates the direct generation of informative samples. We theoretically show how training on challenging, informative examples improves scaling laws and empirically validate that DP achieves better scaling performance with significantly fewer training samples and iterations. On ImageNet-100, DP generates $3.4\times$ fewer samples and requires six times fewer iterations, while on ImageNet-1k, it generates $8\times$ fewer samples with a $30\%$ reduction in iterations, all while achieving superior performance compared to prior work.

## 1. Introduction

A key principle underlying learning in human is deliberate practice (DP)—progress is made not by repeating what is already known but by continuously engaging with tasks that stretch the limits of one's abilities (Ericsson et al., 1993). For example, when learning to play the guitar, simply practicing songs that one has mastered does little to improve skill. Instead, targeted practice on challenging tasks and

refining learning through feedback, leads to real progress. This principle highlights that effective learning requires exposure to informative and difficult examples rather than passive repetition.

In contrast, most machine learning models are trained on pre-collected data that remain static throughout training, limiting their ability to dynamically adapt to their own weaknesses. One promising source of data for visual recognition tasks is large-scale pre-trained text-to-image models (Rombach et al., 2022). They provide an essentially infinite source of synthetic training data, presenting an alternative to real-world datasets, which are often expensive or infeasible to curate (Hemmat et al., 2023; Shin et al., 2023; Zhang et al., 2024). With the great promise of text-to-image models, a natural question arises: what is the potential of learning using **only** synthetic data? Empirical studies show that increasing the volume of synthetic training data often leads to diminishing returns, with performance gains following a power law stagnation (Fan et al., 2024; Tian et al., 2024a). Instead, pruning to remove uninformative examples has proven effective in improving the effectiveness of training with real or synthetic data (Sorscher et al., 2022; Kolossov et al., 2024; Feng et al., 2024).

Inspired by human learning principles and recent advances in generative image models, we propose the Deliberate Practice (DP) for Synthetic Data Generation framework. Unlike static approaches that generate all synthetic training data upfront (Fan et al., 2024; Shin et al., 2023; Hemmat et al., 2023), our framework incorporates a dynamic loop between a diffusion model and a downstream learner throughout the training. More concretely, rather than generating an entire dataset at once and irrespective of the learner and then pruning it to remove uninformative samples, we propose DP to efficiently *generate data directly from the pruned distribution of informative samples*. By leveraging the learner's prediction entropy to guide the generation process, our approach generates only the most challenging and informative training examples.

Our framework operates **dynamically**: we begin with an initial set of synthetic data and train a learner until performance on a real validation set plateaus. At this point, the learner's entropy is used to guide the diffusion model to gen-

---

[*]Equal contribution  [1]FAIR at Meta - Montreal, Paris, and New York City labs [2]Concordia University [3]Mila [4]McGill University [5]Canada CIFAR AI chair. Correspondence to: Reyhane Askari-Hemmat <reyhaneaskari@meta.com>, Mohammad Pezeshki <mpezeshki@meta.com>.

*Proceedings of the 42^{nd} International Conference on Machine Learning*, Vancouver, Canada. PMLR 267, 2025. Copyright 2025 by the author(s).

erate new challenging examples. These examples are added to the training set, and the process repeats, ensuring that the model is continually exposed to increasingly informative data throughout training.

This approach aligns with broader goals in machine learning, such as interactive learning environments, continual learning (Kirkpatrick et al., 2017), and active learning (Settles, 2009). By leveraging a dynamic loop, Deliberate Practice reduces inefficiencies from redundant or already learned data, thereby improving the scaling laws of training with synthetic data.

Our contributions are summarized as:

- We introduce the *Deliberate Practice for Synthetic Data Generation* framework, which dynamically adds new data points when the learner's validation accuracy plateaus [Section 3]. Our framework leverages the learner's prediction entropy to generate **challenging synthetic data**, improving the scaling behavior of synthetic data (Figures 1 and 4).
- We provide a theoretical analysis of the scaling behavior of a simple model trained on selected examples (Section 4). Using random matrix theory, we characterize the test error as a function of data size and the example selection function, showing **improved scaling when prioritizing hard and informative examples**.
- We show that entropy-guided sampling approximates generating from an entropy-pruned distribution (Section 2). We empirically validate that DP can improve the validation accuracy compared to direct pruning while being remarkably **cheaper in compute up to 5×** (Figure 5).
- We demonstrate that DP outperforms prior work on both ImageNet-100 and ImageNet-1k while requiring significantly less data and fewer training iterations. On ImageNet-100, our approach generated **3.4× less samples** and completed training in only one-sixth of the iterations used in prior work, yet still achieved superior performance. Similarly, on ImageNet-1k, we generated **8× less samples** and reduced the number of iterations by 30%, while outperforming previous results (Table 1).
- Furthermore, DP exhibits strong performance on **out-of-distribution** (OOD) datasets, even outperforming models trained with real data on ImageNet-R and ImageNet-Sketch, with **improvements of up to 15%** (Table 1).

## 2. Problem Formulation

**Problem Setup.** Standard supervised learning relies on a large real labeled training set. Here, however, we assume no real training data is available, and instead, we must rely on

a generative model to synthesize training examples.

Formally, let $\mathcal{Y}$ denote the set of class labels. Our goal is to train a classifier $f_\phi : \mathcal{X} \to \mathcal{Y}$, parameterized by $\phi$, which maps inputs $x \in \mathcal{X}$ (*e.g.*, images) to labels $y \in \mathcal{Y}$. We are given a predefined label set $\mathcal{Y}$, a fixed (small) validation set $\mathcal{D}^{\mathrm{val}} = \{(x_i, y_i)\}_{i=1}^n$ consisting of real data for evaluation, and a generative model $g_\theta$ capable of sampling synthetic data conditioned on a label, *i.e.*, $x \sim g_\theta(y)$. However, no real training data is available, *i.e.*, $\mathcal{D}^{\mathrm{tr}} = \varnothing$. The objective is to train $f_\phi$ using *as few generated examples as possible* while maximizing generalization to real data as measured by performance on $\mathcal{D}^{\mathrm{val}}$. The key challenge is to generate minimal yet effective training data, requiring a principled mechanism to select/generate informative examples.

**The Need for Informative Examples.** Not all synthetic samples contribute equally to learning. Prior work shows that simply increasing the synthetic dataset size leads to diminishing returns, as many generated samples are redundant or too easy (Fan et al., 2024). Instead, training should focus on examples that maximize learning efficiency.

Given a measure of *informativeness* for a synthetic sample $x$, one approach is to generate a large dataset and **prune uninformative examples**. Formally, let $\mathcal{D}^{\mathrm{pool}} = \{(x_i, y_i)\}_{i=1}^N$ be a large set of $N$ generated samples. We define a *pruned dataset* as $\mathcal{D}' := \{(x_i, y_i) \mid i \in [N], q_i = 1\}$, where $q_i \in \{0, 1\}$ is a selection variable determining whether a data point $(x_i, y_i) \in \mathcal{D}^{\mathrm{pool}}$ is retained. The subset size is constrained by $m = \sum_{i=1}^N q_i$. The quantity $N/m$ is referred to as the over-sampling ratio.

Let $P$ and $Q$ denote the distributions of the original and pruned datasets, respectively. The pruning process operates as an importance sampling scheme:

$$\mathrm{d}Q = \pi \, \mathrm{d}P, \tag{1}$$

where $\pi$ is a normalized weighting function that retains the informative samples. The generate-then-prune approach ensures that only informative examples are kept, it is **computationally inefficient**, as many generated samples are discarded. This motivates the need to devise mechanisms to directly sample the informative examples.

**Approximate Sampling of Informative Examples.** Suppose that $\mathcal{D}^{\mathrm{pool}}$ is generated using a diffusion model with induced probability $P$. The generative process is governed by a reverse SDE (Song & Ermon, 2019):

$$\mathrm{d}x = \left[v(x, t) - g(t)^2 \nabla \log p_t(x)\right] \mathrm{d}t + g(t) \, \mathrm{d}W(t), \tag{2}$$

where $W(t)$ is a Wiener process, modeling stochastic noise, $v(x, t)$ is a drift term, $g(t)$ is a coefficient controlling the noise level at time $t$, and $\nabla \log p_t(x)$ is the score function.

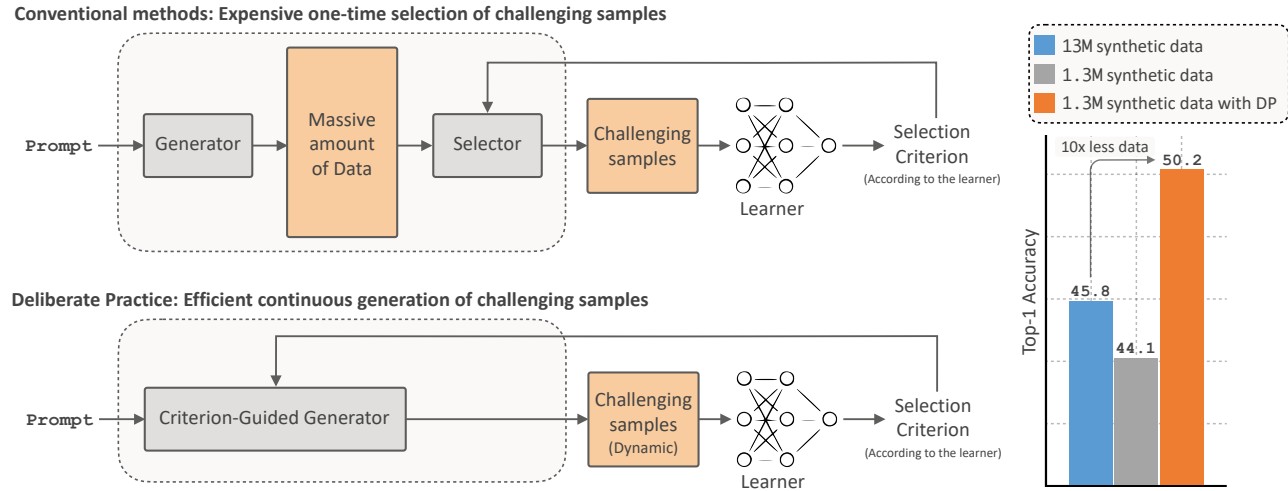

Figure 1: (**Top**): Conventional approaches generate (or collect) a massive static dataset and then select challenging examples in a one-time filtering step based on the learner's selection criterion. This is inefficient, as most generated data is discarded. (**Bottom**): DP **continuously generates only the most challenging examples** based on continuous feedback from the learner, eliminating the need for large-scale data pruning. This iterative process ensures that training focuses on progressively informative examples, improving efficiency and performance. (**Right**): Top-1 validation accuracy on ImageNet-1k with models trained solely on synthetic data. DP (orange) achieves higher accuracy than the 13M synthetic data setup (blue) while using **10× fewer samples**, significantly outperforming the 1.3M baseline (gray).

Instead of sampling from $P$, we aim to sample directly from $Q$ as in Eq. (1). By Girsanov's theorem (Oksendal, 2013), modifying the probability measure from $P$ to $Q$ introduces a correction term in the reverse SDE:

$$
\begin{aligned}
\mathrm{d}x = & \left[ v(x,t) - g(t)^2(\nabla \log p_t(x) + \nabla \log \pi(x,t)) \right] \mathrm{d}t \\
& + g(t)\, \mathrm{d}W(t).
\end{aligned}
\tag{3}
$$

The term $\nabla \log \pi(x,t)$ effectively modifies the score function and biases the sampling distribution according to the weighting function $\pi(x,t)$. This modification allows approximating direct sampling from the pruned distribution $Q$, eliminating the need to first sample uniformly from $P$ and later prune the data.

### 2.1. Efficient Entropy-Guided Sampling with DDIM.

We leverage denoising diffusion implicit models (DDIMs) (Song et al., 2020) for efficient sampling. At each step $t$, the reverse update for generating a conditional sample is:

$$
x_{t-1} = \sqrt{\xi_{t-1}}\hat{x}_{0,t} + \underbrace{\sqrt{1-\xi_{t-1}-\sigma_t^2} \cdot \epsilon_\theta^{(t)}(x_t,y)}_{\text{direction pointing to } x_t} + \underbrace{\sigma_t \epsilon_t}_{\text{random noise}},
$$

where $\epsilon_t$ is random noise and $\sigma_t$ and $\xi_{t-1}$ are time-dependent coefficients. The term $\hat{x}_{0,t}$ approximates the final denoised sample:

$$
\hat{x}_{0,t} = \frac{x_t - \sqrt{1-\xi_t}\epsilon_\theta^{(t)}(x_t,y)}{\sqrt{\xi_t}},
\tag{4}
$$

in which $\epsilon_\theta^{(t)}(x_t,y)$ approximates the conditional score function using a pretrained denoising network (Ho & Salimans, 2022):

$$
\epsilon_\theta(x_t,y) \approx (1+\lambda)\tilde{\epsilon}_\theta(x,y) - \lambda\tilde{\epsilon}_\theta(x)
\tag{5}
$$

where $\lambda$ is called the classifier-free guidance coefficient which controls the strength of conditional sampling on the label.

An efficient way of sampling from a modified diffusion mode as described in Eq. 3 was proposed by Hemmat et al. (2023), where the weighting function is derived from the entropy of the downstream learner, such that,

$$
\log \pi \propto H(f_\phi(x_0)) = -\sum_{y \in \mathcal{Y}} f_\phi(y \mid x_0) \log f_\phi(y \mid x_0).
\tag{6}
$$

To compute the entropy as in Eq. 6, we need the denoised sample $x_0$. The term $\hat{x}_{0,t}$ can be used to cheaply approximate entropy mid-generation. This allows direct sampling of high-entropy examples by modifying the score function:

$$
\tilde{\epsilon}_\theta^{(t)}(x_t,y) = \epsilon_\theta^{(t)}(x_t,y) + \omega\nabla_{x_t}H(f_\phi(\hat{x}_{0,t})),
\tag{7}
$$

where $\omega$ controls the contribution of the entropy-guidance.

In (Hemmat et al., 2023), real data is used to pre-train the learner, enabling an accurate estimation of $\nabla_{x_t}H(f_\phi(\hat{x}_{0,t}))$. However, when real data is unavailable, alternative approaches are needed to assess sample informativeness. In the next section, we propose to leverage the learner itself

**Algorithm 1** Deliberate Practice for Synthetic Data Generation

1: **Input:** Class labels $\mathcal{Y}$, Generative model $g_\theta$, Validation set $\mathcal{D}^{\text{val}}$, Initial dataset size $N$, New data size $P$, Patience $T_{\max}$, Evaluation interval $\tau$.
2: **Output:** Trained classifier $f_\phi$
3: **Initialize:** Generate $\mathcal{D}_0^{\text{tr}}$ with $N$ examples from $g_\theta$. Start training $f_\phi$ with learning-rate warm-up.
4: Set patience counter $T \leftarrow 0$.
5: **while** training **do**
6:     Update $f_\phi$ on a mini-batch drawn uniformly from $\mathcal{D}_k^{\text{tr}}$.
7:     **if** (every $\tau$ iterations) **then**
8:         Evaluate validation accuracy $\mathcal{A}(f_\phi, \mathcal{D}^{\text{val}})$.
9:         Reset $T \leftarrow 0$ if accuracy improves; else increment $T \leftarrow T + 1$.
10:     **end if**
11:     **if** $T \geq T_{\max}$ **then**
12:         Generate $P$ new examples $\mathcal{D}_{\text{new}}$ with feedback:
13:         $\nabla_{z_t} \log p(x_t \mid y) = \nabla_{z_t} \log p_\theta(z_t) + \omega \nabla_{z_t} H(f_\phi(\hat{x}_{0,t}))$
14:         Augment training set: $\mathcal{D}_{k+1}^{\text{tr}} \leftarrow \mathcal{D}_k^{\text{tr}} \cup \mathcal{D}_{\text{new}}$.
15:         Reset $T \leftarrow 0$.
16:     **end if**
17: **end while**
18: **Finalize:** Apply learning rate decay.

during training to evaluate entropy and determine the informativeness of generated samples dynamically.

## 3. The Deliberate Practice Framework for Synthetic Data Generation

In this section, we describe our Deliberate Practice framework, in which we efficiently train the learner with synthetic data in absence of any real data. In particular, we move to a setup where we dynamically expand the dataset throughout the training. Our framework is summarized in Algorithm 1.

**The initial training data.** The framework begins by generating an initial set of $N$ synthetic training examples $\mathcal{D}_0^{tr} = \{(x_i, y_i)\}_{i=1}^N$ using a pre-trained generative model $g_\theta$. For each class $y_i \in \mathcal{Y}$, the generative model samples images $x_i \sim g_\theta(y_i)$ in a class-conditional manner. The classifier $f_\phi$ starts training on this dataset, with a learning-rate warm-up phase.

**Iterative training and additional data.** Training proceeds iteratively with a mechanism to dynamically augment the dataset whenever the classifier's performance stagnates. The process alternates between training the classifier and generating new synthetic examples.

**Patience mechanism.** At regular iteration intervals, $\tau$, the validation accuracy $\mathcal{A}(f_\phi, \mathcal{D}^{\text{val}})$ is evaluated. If no improvement is observed for $T_{\max}$ intervals (patience threshold), the framework triggers new data generation.

**Entropy guided sampling.** When the patience mechanism triggers, $P$ new examples $\mathcal{D}_{\text{new}} = \{(x_j, y_j)\}_{j=1}^P$ are generated. We directly generate samples from the entropy pruned distribution through entropy guided sampling. The entropy is computed based on the current stage of the classifier $f_\phi$. The $\omega$ coefficient controls the effect of entropy-guidance. With $\omega = 0$, we fall back into regular sampling of diffusion models, while $\omega > 0$ results in generations that have a higher entropy under the classifier.

**Training resumption.** The newly generated examples are added to the dataset, $\mathcal{D}_{k+1}^{\text{tr}} = \mathcal{D}_k^{\text{tr}} \cup \mathcal{D}_{\text{new}}$. After augmenting the dataset, training resumes with a constant learning rate until the patience mechanism is triggered again. Mini-batches are drawn uniformly from the updated pool, which grows dynamically from size $N$ to $N + kP$ after $k$ iterations of augmentation. This cycle is continued until we reach the cool-down phase where the learning rate is decreased and no more new data is added. See Figure 2 for training dynamics of a classifier training with DP.

In Section 4, we provide an intuitive theoretical framework to study the scaling behavior of a simplified DP. In Section 5, we validate the effectiveness of DP in large-scale experiments.

## 4. Training on Informative Examples Improves the Scaling Laws

Before presenting empirical results, we first analyze how selecting informative examples affects the scaling of synthetic data. We study a high-dimensional linear classifier trained with uniform vs. selective sampling and derive an analytic expression for test error using random matrix theory (RMT). Our results show that selecting hard examples improves scaling laws, providing theoretical justification for our approach.

### 4.1. Theoretical Analysis under an Idealized Setup.

Consider a simple generative model for training data:

$$x \sim \mathcal{N}(0, \Sigma), \quad y = \text{sign}(w_0^\top x), \tag{8}$$

where $w_0 \in \mathbb{R}^d$ is the ground-truth labeling function. This gives a distribution $P$ on $\mathbb{R}^d \times \mathbb{R}$.

We study the impact of *uniform sampling* versus *selective sampling* of informative examples on generalization. To formalize this, we assume a pool of $n$ i.i.d. training pairs:

$$X \in \mathbb{R}^{n \times d}, \quad Y \in \mathbb{R}^n. \tag{9}$$

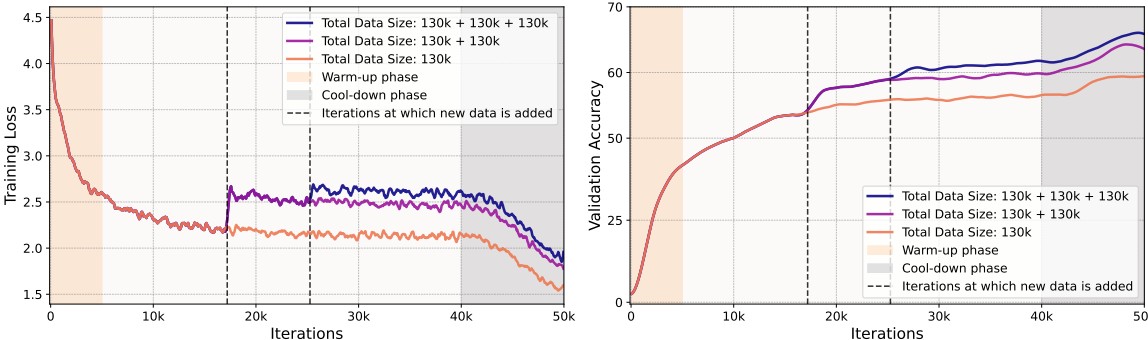

Figure 2: Training loss (left) and validation accuracy (right) of Deliberate Practice on ImageNet-100. The classifier begins training on an initial static dataset (130k samples) until validation accuracy plateaus. At this point, additional samples are generated using entropy-guided sampling, focusing on hard/informative examples. The two dashed vertical lines indicate points where new data is added. We compare three setups: (1) Orange: No additional data is added, training only on the initial dataset. (2) Purple: One round of entropy-guided data generation adds 130k samples. (3) Blue: Two rounds of entropy-guided data generation, adding 260k samples in total. Each data addition leads to an accuracy boost, demonstrating the effectiveness of DP in improving performance with fewer training iterations. For clarity, this figure shows only two rounds of data addition, but in practice, more rounds occur based on the allowed maximum patience. Notably, while training loss increases with new data, validation accuracy steadily improves, showing that the model benefits from progressively challenging examples, ultimately reducing the generalization gap.

A linear classifier $\hat{w}$ is trained using the following loss:

$$\hat{w} = \arg\min_w \quad \frac{1}{n}\sum_{i=1}^n q_i \ell(w^\top x_i, y_i) + \frac{\lambda}{2}\|w\|^2. \quad (10)$$

where $\ell(z, y) = (z - y)^2/2$ is the squared loss, $\lambda > 0$ is a regularization parameter, and $q_i := q(x_i^\top w_s)$ is a selection strategy that determines whether an example is included in training based on its projection in a given direction $w_s \in \mathbb{R}^d$, and an arbitrary measurable binary function $q : \mathbb{R} \to \{0, 1\}$ which encodes the selection strategy.

The *selection/pruning ratio* is given by:

$$p = \mathbb{E}[q(x^\top w_s)] \text{ for } x \sim \mathcal{N}(0, \Sigma). \quad (11)$$

The resulting classifier has a closed-form solution:

$$\hat{w} = \frac{1}{n} R X^\top D Y, \quad R := \left(\frac{1}{n} X^\top D X + \lambda I_d\right)^{-1}, \quad (12)$$

where $D \in \mathbb{R}^{n \times n}$ is a diagonal matrix with $D_{ii} = q_i$.

Our objective is to analyze the asymptotic test error of $\hat{w}$:

$$E_{test}(\hat{w}) = \mathbb{P}(\text{sign}(x^\top \hat{w}) \neq y), \quad (13)$$

where $(x, y)$ is a test example,

### 4.2. Asymptotic Behavior of the Test Error.

We leverage random matrix theory (RMT) techniques (Couillet & Liao, 2022; Liao & Mahoney, 2021; Firdoussi

et al., 2024) to characterize the test error in Eq. (13). Our analysis is based on the spectral density of the resolvent matrix $R$ in Eq. (12), allowing us to compute the first two moments of $yx^\top \hat{w}$ for a test sample $x$ and derive an expression for the test error. For simplicity, we assume an isotropic setup where $\Sigma = I_d$ and defer the general case to Appendix A.

We shall work in the following so-called high-dimensional proportionate scaling regime

$$d, n \to \infty, \quad d/n \to \phi, \quad (14)$$

in which the input-dimension $d$ and the sample size $n$ diverge to infinity at the same rate. The scalar $\phi \in (0, \infty)$ captures the effective dimensionality or over-parametrization rate of the problem.

**Key Scalars.** WLOG, assume $\|w_s\| = 1$. It turns out that the for fixed, pruning, $p$, the asymptotic test error is fully captured by the following scalars:

$$\rho := w_s^\top w_0/\|w_0\|, \ \tau := \frac{\rho}{\sqrt{1 - \rho^2}}, \ \gamma := \mathbb{E}[q(G)G^2],$$

$$\beta := 2\mathbb{E}[q(G)\varphi(\tau G)], \quad \tilde{\beta} := 2\mathbb{E}[q(G)\Phi(\tau G)G], \quad (15)$$

where $G \sim \mathcal{N}(0, 1)$ with pdf $\varphi$ and cdf $\Phi$. Note that $\rho$ quantifies the alignment between the pruning direction $w_s$ and the ground-truth labeler $w_0$, while $\beta$ and $\gamma$ capture statistical properties of the pruning strategy $q$.

**Spectral functions.** The Stieltjes transform $m$ of the limiting spectral density of the resolvent matrix $R$ is shown in

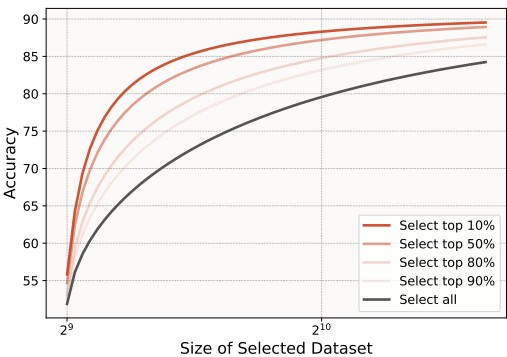

Figure 3: Theoretical prediction for scaling behavior of accuracy (Theorem 1) for a simple classifier in a $d = 512$ dimensional input space, as a function of dataset selection strategy. The classifier is trained on synthetic data with different pruning probabilities, where higher pruning probability corresponds to keeping only the most challenging examples (those closer to the decision boundary).

Lemma 3 to be given by the exact formula (with $z := -\lambda$)

$$m(z) = \frac{p - \phi - z - \sqrt{(p - \phi - z)^2 - 4\phi z}}{2\phi z}, \quad (16)$$

and will play an important role in our theory. The above formula represents a somewhat distorted Marchenko-Pastur law. Indeed, the classical MP (Marčenko & Pastur, 1967) corresponds to $p \to 1$ (i.e. no data pruning).

We further define the following auxiliary functions:

$$
\begin{aligned}
s(z) &:= \frac{\gamma}{1 + \phi m(z)}, \quad \tilde{m}(z) := \frac{1}{s(z) - z}, \\
r(z) &:= \omega^2 \cdot m(z) + \tilde{\omega}^2 \cdot \tilde{m}(z), \\
\text{with } \omega &:= \sqrt{1 - \rho^2}\beta, \quad \tilde{\omega} := \rho\tilde{\beta}.
\end{aligned}
\quad (17)
$$

**Main Result: Test Error Scaling w.r.t Selection Strategy.**

**Theorem 1.** *In the limit Eq. (14), the classification test error satisfies:* $E_{test}(\hat{w}) \to \arccos\left(|m_0|/\sqrt{\nu_0}\right)/\pi$, *where*

$$
\begin{aligned}
m_0 &:= \omega m(-\lambda) + \tilde{\omega}\tilde{m}(-\lambda), \\
\nu_0 &:= p\phi m'(-\lambda) + r'(-\lambda) - \frac{2\phi m'(-\lambda)}{1 + \phi m(-\lambda)r(-\lambda)}.
\end{aligned}
$$

The scaling behavior of test error is fully determined by the six scalars $(\lambda, \phi, p, \rho, \gamma, \beta, \tilde{\beta})$. Importantly, the choice of the data point selection strategy $i \mapsto q(x_i^\top w_s)$ only influences performance through $\rho, \gamma, \beta$, and $\tilde{\beta}$.

### 4.2.1. EXAMPLE: SELECTING INFORMATIVE EXAMPLES.

Consider a selection function of the form $q_i = q(x_i^\top w_s)$ for all $i$, where,

$$q(t) := 1[|t| \le \xi] = \begin{cases} 1, & \text{if } |t| \le \xi, \\ 0, & \text{else}, \end{cases} \quad (18)$$

for some threshold $\xi \ge 0$. Such selection strategy selects only the examples near the decision boundary of $w_s$, analogous to using classifier entropy as a selection criterion but simpler to study. Lemma 1 and 2 derive explicit expressions for $(\gamma, \beta, \tilde{\beta})$. Figure 3 presents theoretical predictions for test accuracy across different degrees of example selection, showing that *selecting hard examples improves scaling laws*, reducing the number of training samples needed for the same performance. However, beyond a certain point, excessive pruning degrades performance, as illustrated in Figure 5.

### 4.2.2. ADAPTIVE SELECTION STRATEGY.

Data selection relies on a pruning direction $w_s$ to select informative/hard examples: $i \mapsto q(x_i^\top w_s) \in \{0, 1\}$, but these examples are ultimately used to train $\hat{w}$. If $w_s$ and $\hat{w}$ are misaligned, what is considered hard by $w_s$ may not be hard for $\hat{w}$, reducing the effectiveness of selective sampling. In fact, hard examples change over time: an example that was identified hard, might not remain hard are more training is done. To ensure alignment, $w_s$ should periodically update to reflect the evolving decision boundary of $\hat{w}$. This adaptive selection mechanism motivates the continuous data generation process of DP, as presented in Section 3.

Data selection relies on a pruning direction $w_s$ to identify informative or hard examples: $i \mapsto q(x_i^\top w_s) \in \{0, 1\}$. However, these selected examples are ultimately used to train $\hat{w}$, and if $w_s$ and $\hat{w}$ are misaligned, what is considered hard by $w_s$ may not be hard for $\hat{w}$, reducing the effectiveness of selective sampling. In fact, $w_s$ and $\hat{w}$ deviate from each other the more $\hat{w}$ is trained on these examples. Moreover, the definition of "hard" changes over time—an example that was initially difficult may become easier as training progresses. To maintain alignment, $w_s$ should be periodically updated to reflect the evolving decision boundary of $\hat{w}$. This adaptive selection mechanism underpins the continuous data generation process in DP, as presented in Section 3.

## 5. Experiments

For all the experiments, we use the LDM1.5 (Rombach et al., 2022) as the pre-trained text-to-image (T2I) model. We studied four different T2I models and found this model outperforming the rest. For more details see Appendix D.1.

**Datasets.** We validate our framework on two datasets.

ImageNet-100 (Tian et al., 2020; Sarıyıldız et al., 2023), a subset of ImageNet-1k (Deng et al., 2009), containing 100 classes and 5k validation examples, where the real validation set is used for evaluation and the real training set (126,689 examples) serves as a held-out test set. We also conduct experiment ImageNet-1k, using the 50k validation examples to monitor performance and reserving the real training set (1.3 million examples) as a held-out test set.

## 5.1. Scaling Laws of Synthetic Data

We train a Vision Transformer (ViT-B) (Dosovitskiy et al., 2021) classifier with synthetic data. We study two scenarios: 1) Static data generation and 2) Deliberate Practice (DP). In all the experiments in this section we have a fixed and controlled setup. We train the models for 100k and 50k iterations for ImageNet-1k and ImageNet-100 respectively. For additional details, see Appendix D.5.

**Static data generation.** In this setup, all data is generated before training, and the classifier is trained on a fixed dataset. We experiment with different dataset sizes to see its impact on accuracy.

**Deliberate Practice data generation.** Hyperparameters $\omega$ and $\lambda$ are tuned on ImageNet-100 and found effective for ImageNet-1k as well (see Section D.5 for details). We track validation accuracy throughout training and use it to determine when to generate new data, following a patience-based criterion. To ensure the model has not over-fitted to the validation set, we also report accuracy on the full real training sets of ImageNet-100 and ImageNet-1k, used as *held-out* test sets.

Figure 4 compares the scaling laws of the **Static** and **Deliberate Practice (DP)** on ImageNet-100 and ImageNet-1k. On both datasets, we note that DP scales well with dataset size and it consistently outperforms the Static setup, achieving higher validation accuracy at any given dataset size. On ImageNet-100 we observe that DP can reach the best accuracy of the static setup (with 3 million examples) using only 400k examples. This means that DP requires $7.5\times$ less data to reach the same performance. On ImageNet-1k, we observe that DP can outperform the best accuracy of the static setup (with 13 million examples), using only 640k examples. This translates to DP requiring $20\times$ less data to outperform the Static setup. For additional details on the hyper-parameters of these experiments, see Appendix D.5.1. Refer to Figure 13 for a visualization of how the dataset evolves from the start to the end of training.

## 5.2. Comparison with Previous Work

We compare DP with prior works on synthetic data generation for image classification (Sarıyıldız et al., 2023; Fan et al., 2024). Specifically, we evaluate setups that use class-

names for prompting and publicly available models for sample generation. Performance is assessed on real ImageNet (held-out) training and validation sets, as well as on ImageNet-V2 (Recht et al., 2019), ImageNet-Sketch (Wang et al., 2019), ImageNet-R (Hendrycks et al., 2021a), and ImageNet-A (Hendrycks et al., 2021b) to measure out-of-distribution (OOD) generalization.

The results in Table 1 show that DP outperforms prior benchmarks on both ImageNet-100 and ImageNet-1k while requiring significantly less data and fewer training iterations. On ImageNet-100, DP generated 4.6 million fewer samples and trained for only one-sixth of the iterations compared to previous works, yet achieved superior performance on the real data. Similarly, on ImageNet-1k, DP reduced sample generation by 56.2 million and cut training iterations by over 30%, while still outperforming previous results.

Furthermore, models trained with DP exhibit strong performance on out-of-distribution datasets, even surpassing models trained on real data on ImageNet-R and ImageNet-Sketch, with improvements of up to 15%.

## 5.3. Connection Between Pruning and DP

In Section 2, we discussed how DP approximates direct sampling from a pruned distribution. Here, we validate this experimentally on ImageNet-100 using two setups:

1. **Oversampling then Pruning:** Generate a large pool and select high-entropy samples.

2. **Direct entropy-guided generation:** Generate only informative samples (a special case of DP with a single step of data addition).

We start with 130k generated samples (regular vanilla sampling), train for 17k iterations, then add a one-time additional 130k samples, increasing the total data size to 260k and training for an additional 33k iterations.

In setup 1, we vary the pool size, ranging from no pruning (130k pool) up to an oversampling ratio of 18 (2.4M pool), selecting the top 130k high-entropy samples. In setup 2, we generate exactly 130k entropy-guided samples, varying the entropy-gauidance coefficient.

Figure 5 (a, b) shows that both methods improve performance up to a point, after which excessive selection of high-entropy samples leads to degradation—likely due to selecting high-entropy but harmful outliers. This aligns with our theoretical predictions in Figure 5 (c).

Regarding computational costs, generating a single image with entropy-guidance on an Nvidia H100 takes $1.82\times$ longer than standard vanilla sampling. However, achieving similar performance through oversampling requires significantly more data, leading to a linear increase in cost.

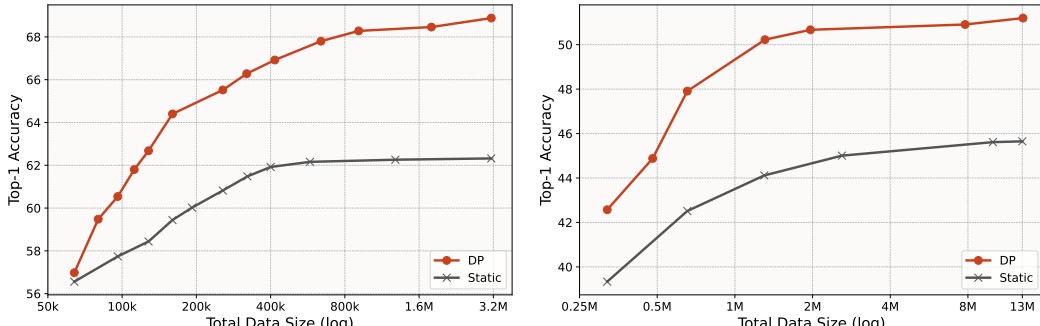

Figure 4: **Scaling laws of synthetic data.** Real Validation accuracy versus total dataset size for the Static (pink ×), and Deliberate Practice (blue o) setups on ImageNet-100 (left) and ImageNet-1k (right). DP significantly outperforms Static data generation, achieving higher accuracy with fewer synthetic examples. DP achieves the same accuracy as the static setup using 7.5× less data on ImageNet-100 and 20× less data while outperforming it on ImageNet-1K.

Table 1: **Comparison with previous work.** DP outperforms other models on both ImageNet-100 and ImageNet-1k while requiring significantly less data and fewer training iterations. Note that DP experiments reported in this table are trained longer than models reported in the previous section and, consistent with other work, use a smaller classifier free guidance scale of $\lambda = 2$.

|  | Task | # Iters | Data size | IN real Val. | IN real tr. | IN-v2 | IN-Sk | IN-R | IN-A |
|---|---|---|---|---|---|---|---|---|---|
| Real | IN-100 | 100k | 130k | 88.5 | - | **76.4** | 37.1 | 60.8 | **33.5** |
| Syn. Static - (Sarıyıldız et al., 2023) | IN-100 | 13k | 130k | 63.5 | - | 62.7 | 41.8 | 64.2 | 13.7 |
| Syn. Static - (Sarıyıldız et al., 2023) | IN-100 | 635k | 6.5M | 73.3 | - | 72.3 | 42.0 | 59.4 | 17.1 |
| Syn. DP (ours) | IN-100 | 100k | 1.9M | **74.3** | 75.0 | 66.3 | **52.0** | **76.6** | 25.9 |
| Real | IN-1k | 200k | 1.3M | 82.6 | - | **70.9** | 32.5 | 44.6 | **29.4** |
| Syn. Static - (Sarıyıldız et al., 2023) | IN-1k | 130k | 1.3M | 42.9 | - | 43.0 | 16.6 | 26.3 | 3.6 |
| Syn. Static - (Fan et al., 2024) | IN-1k | 210k | 2M | 50 | - | 42.2 | 27.2 | 45.7 | 6.6 |
| Syn. Static - (Fan et al., 2024) | IN-1k | 315k | 64M | 54 | - | 46.0 | 32.4 | 52.5 | 9.4 |
| Syn. DP (ours) | IN-1k | 200k | 6.5M | 54.1 | 54.84 | 48.5 | 34.7 | 56.0 | 12.3 |
| Syn. DP (ours) | IN-1k | 200k | 9.1M | **55.1** | 55.73 | 49.3 | **36.0** | **57.2** | 13.4 |

As a result, DP is 5× more efficient while also providing higher absolute improvements compared to pruning-based selection. See Figure 5 for details and Figure 11 for some visualizations.

### 5.4. The Evolution of Hard Examples Over Time

"Does the sample hardness change as training progresses?"

To answer this question, Figure 6 tracks the error on examples that were misclassified at the time they were added. As expected, once introduced, the model gradually learns to classify them correctly. However, an interesting trend emerges: even before these examples were added, their error was lower than at the moment of inclusion. This suggests that the notion of hardness is dynamic—what is considered challenging at one point may become easier over time. Conversely, examples that were once easy might later become difficult due to shifts in the learned decision boundaries. This highlights a key limitation of static pruning approaches and underscores the importance of dynamically adapting

the selection of informative examples throughout training, as done in Deliberate Practice (DP). See Figure 12 for some visualization of generations through training.

## 6. Related Work

**Synthetic data for training neural networks.** Synthetic data has become a powerful tool for training machine learning models across various domains. For instance, text-to-image diffusion models have been successfully used for visual representation learning (Astolfi et al., 2023; Li et al., 2025; Tian et al., 2024a;b; Sarıyıldız et al., 2023). However, limitations of synthetic data are highlighted by Fan et al. (2024), emphasizing the importance of generating more challenging and informative examples. Addressing distribution shifts between synthetic and real data, Hemmat et al. (2023) and Yuan et al. (2023) propose synthesizing training data that matches real data distributions or conditioning on real examples to reduce this gap. Expanding small-scale datasets has also been studied, see e.g. Zhang

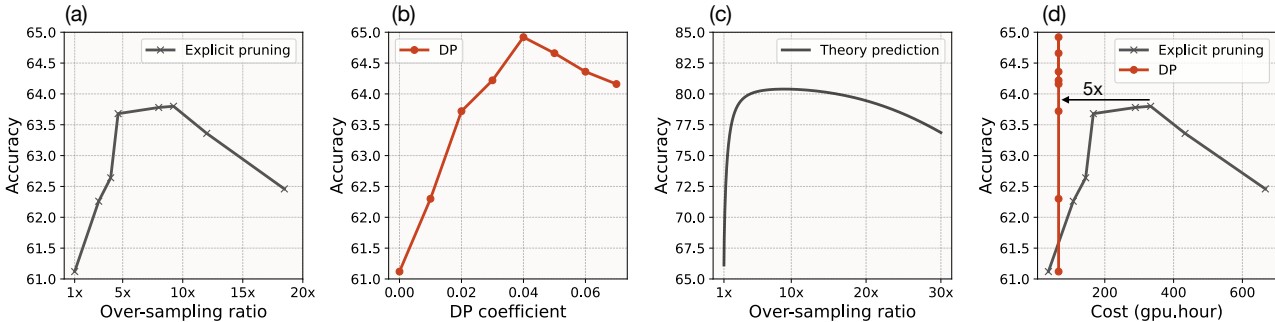

Figure 5: Plots describing the performance of DP compared to explicit pruning and theory prediction while changing the oversampling ratio or the DP coefficient. (a) Over-sampling with entropy-based selection – Generate a large pool of samples (ranging from 130k to 2.4M) and select the 130k highest-entropy examples. (b) Generate 130k high-entropy examples directly using DP with varying entropy guidance strength through $\omega$. (c) The theory prediction on the accuracy based on the over-sampling ration. (d) Comparing the compute cost of DP vs oversampling then pruning. We observe that DP exhibits a similar accuracy curve compared to explicit pruning and theoretical prediction when changing the over-sampling/DP coefficient. However, DP is computationally remarkably more efficient while gaining more accuracy delta.

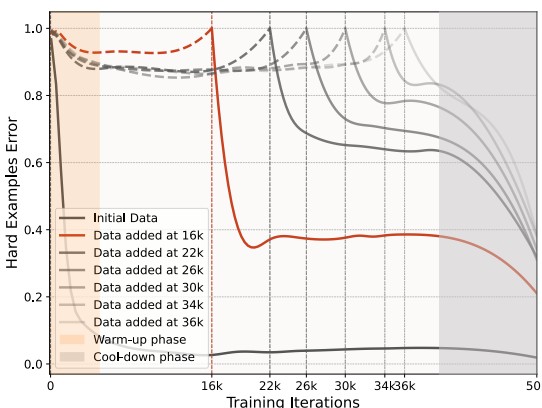

Figure 6: Error trajectories of hard (misclassified) examples added at different training stages. The red curve highlights the first batch of added data for better visibility, but the same trend applies to all batches. Notably, even before being trained on, these examples exhibit a lower error rate than at their point of inclusion, indicating that hardness is not static, it evolves throughout training.

et al. (2024). Another related line of work involves using VLMs and LLMs to generate descriptions for augmenting datasets (Dunlap et al., 2023).

Synthetic data is increasingly used to train (LLMs). For example, LLaMA3 (Grattafiori et al., 2024) employs AI-generated data for fine-tuning. Similarly, self-play approaches, e.g., Yuan et al. (2024), align with our framework by generating increasingly difficult examples for training.

**Continual learning and active learning.** Our work is also closely related to principles from active learning (Bang et al., 2024; Evans et al., 2023) and continual learning, which

prioritize iterative model updates with tailored data. These methods highlight the importance of selecting informative samples based on the model's current state. (Sorscher et al., 2022) showed that pruning static datasets using metrics like margin scores can improve scaling laws by retaining the most informative examples, albeit in a non-adaptive manner.

**Challenges and risks of synthetic data.** The challenges of training models on synthetic data, have gained significant attention. Dohmatob et al. (2024a;b) studied "model collapse", a phenomenon where iterative training on synthetic data degrades performance. They emphasize that data verification mechanisms can mitigate this risk and enable scaling with synthetic data. Similarly, our framework by generating informative examples through a dynamic loop, improves sample efficiency.

## 7. Conclusion

We introduced Deliberate Practice for Synthetic Data Generation, a framework that improves scaling laws by dynamically generating challenging and informative training examples. Unlike traditional methods that rely on static datasets, our approach approximates generating data directly from a pruned distribution, reducing inefficiencies and ensuring models continuously training on informative samples. We provided theoretical insights into the benefits of training on pruned distributions and empirically demonstrated that our method significantly improves performance while requiring fewer training iterations. Our results on ImageNet-100 and ImageNet-1K show that Deliberate Practice achieves superior accuracy with far less data and compute, outperforming previous state-of-the-art. Our work highlights the potential of structured synthetic data generation in advancing efficient and adaptive learning.

## Impact Statement

This work introduces a method for improving the sample efficiency of synthetic data generation through a deliberate practice framework that prioritizes the most informative training examples. By enabling more efficient use of generated data, our approach may reduce the computational and environmental costs associated with training large models, and make data generation more accessible in low-resource settings.

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

# A. Further Theoretical Analysis and proofs

## A.1. The Unregularized Regime

We now consider our theory in the limit $\lambda \to 0^+$. Thus, the parameter vector for the classifier is the least-squares estimate for $w_0$, i.e $\hat{w} = \hat{w}_{LS} = X'^{\dagger}Y'$. Recall the definition of the constants $\gamma$, $\beta$, $\tilde{\beta}$, $\omega$, and $\tilde{\omega}$ from equations (17). Recall that $p \in (0, 1]$ is the proportion of training data left after pruning the original dataset $(X, Y)$ containing $n$ examples. We have the following important corollary to Theorem 1.

**Corollary 1.** *In the (ordered) limit $n, d \to \infty$, $d/n \to \phi$, $\lambda \to 0^+$, it holds that $E_{test}(\hat{w}) \to \arccos(|a|/\sqrt{b})/\pi$, where the constants $a$ and $b$ are given as follows:*

*(A) If $\phi < p$, then*

$$a := (\omega + \tilde{\omega}p/\gamma)/(p - \phi), \qquad b := \frac{p^2\phi + (r'_0 - 2\phi r_0)}{(p - \phi)^3}, \tag{19}$$

$$\text{with } r_0 := \beta^2 + \tilde{\beta}^2 p/\gamma, \quad r'_0 := p \cdot \left(\beta^2 + \tilde{\beta}^2 \cdot ((p - \phi)p/\gamma^2 + \phi/\gamma)\right). \tag{20}$$

*(B) If $\phi > p$, then*

$$a := (\omega + \tilde{\omega}/c_1)c_0, \quad b := (p\phi - r_0)c_0, \tag{21}$$

$$\text{with } c_0 := 1 - p/\phi, \quad c_1 := 1 - (p - \gamma)/\phi, \quad r_0 := \beta^2 + \tilde{\beta}^2/c_1. \tag{22}$$

The result is empirically verified in Figure 7(a).

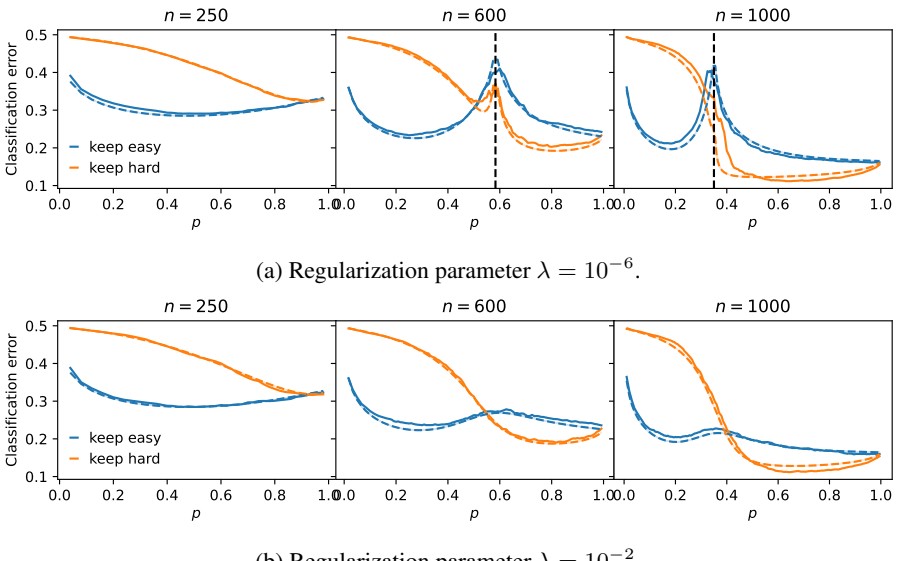

(a) Regularization parameter $\lambda = 10^{-6}$.

(b) Regularization parameter $\lambda = 10^{-2}$.

Figure 7: **Empirical verification of Theorem 1 and Corollary 1.** For this experiment, the input dimension is $d = 350$, and each subplot corresponds to a different value of the original sample size $n$. The experiment for $\lambda = 10^{-6}$ is a proxy for the unregularized case $\lambda \to 0^+$. Solid lines correspond to observed values of the test error $E_{\text{test}}(\hat{w})$, while broken lines are the theoretical prediction of Theorem 1 (**bottom row**) and Corollary 1 (**top row**). Notice the excellent match between the experimental results and our theory. Also, observe the multiple-descent patterns, reminiscent of a non-trivial effect of different pruning strategies in different regimes of the pruned training dataset size $n_0 = np$; the vertical line corresponds to an interpolation threshold at $p = \phi$, i.e., $n_0 = d$.

## A.2. Some Important Examples of Pruning Strategies

**Keep Hard Examples (KH).** Consider the case where the pruning strategy is given by $q_i = q_{KH}(x_i^\top w_s)$ for all $i$, where

$$q_{KH}(t) := 1[|t| \le \xi] = \begin{cases} 1, & \text{if } |t| \le \xi, \\ 0, & \text{else,} \end{cases} \tag{23}$$

for some $\xi \ge 0$. Define $\alpha := \xi/\|w_s\|$. We have explicit formula for the constants $\beta$ and $\tilde{\beta}$ appearing in Theorem 1. Viz,

**Lemma 1.** *With* $\tau := \rho/\sqrt{1-\rho^2}$, $\epsilon_1 := 2\Phi(\alpha/\sqrt{1-\rho^2}) - 1$, *and* $\epsilon_2 := 2\Phi(\tau\alpha) - 1$, *it holds that*

$$\tilde{\beta}(q_{KH}) = 2(\rho\varphi(0)\epsilon_1 - \varphi(\alpha)\epsilon_2), \quad \beta(q_{KH}) = 2\varphi(0)\sqrt{1-\rho^2} \cdot \epsilon_1. \tag{24}$$

**Example 2: Keep Easy Examples (KE).** Here, the pruning strategy is $q_i = q_{KE}(x_i^\top w_s)$, where

$$q_{KE}(t) := 1[|t| > \xi] = \begin{cases} 0, & \text{if } |t| \le \xi, \\ 1, & \text{else.} \end{cases} \tag{25}$$

**Lemma 2.** *With* $\tau := \rho/\sqrt{1-\rho^2}$, $\epsilon_1 := 2(1 - \Phi(\alpha/\sqrt{1-\rho^2}))$, $\epsilon_2 := 2\Phi(\tau\alpha) - 1$, *it holds that*

$$\tilde{\beta}(q_{KE}) = 2(\rho\varphi(0)\epsilon_1 + \varphi(\alpha)\epsilon_2), \quad \beta(q_{KE}) = 2\varphi(0)\sqrt{1-\rho^2} \cdot \epsilon_1. \tag{26}$$

**Example 3: Interpolation between Keep Hard and Keep Easy Strategies.** Consider the following pruning strategy proposed in (Kolossov et al., 2024)

$$q(t) \propto \sigma(t)^\omega (1 - \sigma(t))^\omega, \tag{27}$$

for some tuning parameter $\omega$. Here, $\sigma$ is the sigmoid function. We can associate $q(x_i^\top w_s)$ with the probability the auxiliary classifier $x \mapsto sign(x^\top w_s)$ assigns to an example $x_i$. Thus, positive values of $\omega$ correspond to keeping examples considered uncertain (i.e hard) by this classifier, while negative values correspond to examples considered easy.

## A.3. Main Ingredients of Proofs

### A.3.1. DETERMINISTIC EQUIVALENT FOR THE RESOLVENT MATRIX $R$

**Definition 1** (Deterministic Equivalents). *Given a sequence of random $N \times N$ matrices $(R_N)_N$, a deterministic equivalent thereof is a sequence of deterministic $N \times N$ matrices $(\overline{R}_N)_N$ such that*

$$\operatorname{tr} A_N (R_N - \overline{R}_N) \overset{a.s}{\to} 0, \tag{28}$$

*for all sequences of $N \times N$ matrices $(A_N)_N$ with bounded Frobenious norm.*

Let $\Pi$ (resp. $\Pi_\perp = I_d - \Pi$) be the projection onto the span (resp. orthogonal complement of the span) of $w_s$. Define the following auxiliary vectors and scalars

$$v = \Sigma^{1/2} w_s, \quad v_1 = \frac{v^\top w_s}{\|w_s\|}, \quad v_\perp = \Pi_\perp v. \tag{29}$$

Note that $v_\perp$ is $(d-1)$-dimensional and $\|v_\perp\| = \sqrt{\|v\|^2 - v_1^2}$.

Henceforth we make the replacement $z = -\lambda < 0$, so that the resolvent matrix $R$ now writes

$$R = R(z) := (X^\top D X/n - zI_d)^{-1}. \tag{30}$$

Let $\delta(z)$ be the unique positive solution to the fixed-point equation

$$m(z) = d^{-1} \operatorname{tr} \bar{R}_b(z), \quad \delta(z) = n^{-1} \operatorname{tr} \Sigma \bar{R}_b(z), \tag{31}$$

$$\bar{R}_b(z) = \left( \mathbb{E}_{x \sim \mathcal{N}(0,\Sigma)} \left[ \frac{q(x^\top w_s)}{1 + q(x^\top w_s)\delta(z)} \right] \Sigma - zI_d \right)^{-1}. \tag{32}$$

Note that the inner expectation evaluates to

$$\mathbb{E}_{x \sim \mathcal{N}(0,\Sigma)} \left[ \frac{q(x^\top w_s)}{1 + q(x^\top w_s)\delta(z)} \right] = \frac{p}{1 + \delta(z)} =: t(z),$$

and so $\bar{R}_b(z) = (t(z)\Sigma - zI_d)^{-1}$. Observe that $\bar{R}_b(z)(t(z)\Sigma - zI_d) = I_d$, and so $t(z)\Sigma\bar{R}_b(z) = I_d + z\bar{R}_b(z)$. We deduce that

$$t(z)\delta(z) = n^{-1} \operatorname{tr} t(z)\Sigma\bar{R}_b(z) = n^{-1} \operatorname{tr}(I_d + z\bar{R}_b(z)) = \phi \cdot (1 + zm(z)).$$

Thus the equations defining $m(z)$ and $\delta(z)$ can be rewritten as

$$m(z) = d^{-1} \operatorname{tr}(t(z)\Sigma - zI_d)^{-1}, \tag{33}$$

$$t(z) = \frac{p}{1 + \delta(z)}, \tag{34}$$

$$\phi \cdot (1 + zm(z)) = t(z)\delta(z) = t(z)\left( \frac{p}{t(z)} - 1 \right) = p - t(z). \tag{35}$$

Solving for $\phi zm(z)$ in terms of $t(z)$ in the last equation gives

$$\phi zm(z) = \frac{p\delta(z)}{1 + \delta(z)} - \phi = p - \phi - \frac{p}{1 + \delta(z)} = p - \phi - t(z).$$

Plugging this into the first equation gives the following fixed-point equation for $t(z)$

$$p - \phi - t(z) = zn^{-1} \operatorname{tr}(t(z)\Sigma - zI_d)^{-1}. \tag{36}$$

The following result shows that $\bar{R}$ is a deterministic equivalent for $R$.

**Proposition 1.** *Recall the function $t(z)$ as the unique positive solution to the equation* (36). *Then,*

$$R \simeq \bar{R}, \ with \ \bar{R} = \Sigma^{-1/2}(\bar{m}(z)\Pi_\perp + \tilde{m}(z)\Pi)\Sigma^{-1/2}, \tag{37}$$

$$where \ \bar{m}(z) = \frac{1}{t(z) - z}, \quad \tilde{m}(z) = \frac{1}{s(z) - z}, \quad s(z) = \frac{\gamma}{1 + \delta(z)} = (\gamma/p)t(z), \tag{38}$$

$$\gamma := \mathbb{E}[q(G)G^2], \ for \ G \sim \mathcal{N}(0,1). \tag{39}$$

## A.4. Isotropic Case

Consider the special case where the covariance matrix is $\Sigma = I_d$. It is not hard to see that we must have $\bar{m}(z) \equiv m(z) \equiv \delta(z)/\phi$. Let us now compute $m(z)$.

**Lemma 3.** *For every $z = -\lambda < 0$, $m(z)$ is given by formula* (16).

*Proof.* Indeed, observe that in the isotropic case the equation (36) reduces to $p - \phi - t(z) = \phi z/(t(z) - z)$, or equivalently

$$0 = \phi z + (t(z) - p + \phi)(t(z) - z) = t(z)^2 - (p - \phi + z)t(z) + pz.$$

The discriminant of this quadratic equation evaluates to

$$(p - \phi + z)^2 - 4pz = (p - \phi - z + 2z)^2 - 4pz$$
$$= (p - \phi - z)^2 + 4z^2 + 4z(p - \phi - z) - 4pz$$
$$= (p - \phi - z)^2 - 4\phi z,$$

and so because $z = -\lambda < 0$, the positive solution is

$$t(z) = \frac{p - \phi + z + \sqrt{(p - \phi - z)^2 - 4\phi z}}{2}.$$ (40)

We deduce that

$$m(z) = \frac{1}{t(z) - z} = \left( \frac{p - \phi - z + \sqrt{(p - \phi - z)^2 - 4\phi z}}{2} \right)^{-1}$$

$$= 2 \cdot \frac{p - \phi - z - \sqrt{(p - \phi - z)^2 - 4\phi z}}{(p - \phi - z) - ((p - \phi - z)^2 - 4\phi z)}$$

$$= \frac{p - \phi - z - \sqrt{(p - \phi - z)^2 - 4\phi z}}{2\phi z},$$

which is precisely the claimed formula given in (16). $\square$

The following result then follows directly from Proposition 1.

**Corollary 2.** *In the isotropic setting, we have the following deterministic equivalents:*

$$R \simeq \bar{R}, \text{ with } \bar{R} = m(z)\Pi_\perp + s(z)\Pi,$$ (41)

$$R^2 \simeq m'(z)\Pi_\perp + \tilde{m}'(z)\Pi.$$ (42)

*where* $\tilde{m}(z) := 1/(s(z) - z)$, $s(z) = \gamma/(1 + \phi m(z))$, *and* $\gamma \geq 0$ *is as given in* (43).

$$\rho = \frac{w_s^\top w_0}{\|w_s\|\|w_0\|}, \ \beta := \mathbb{E}\left[q(\|w_s\|G_2)|G_1|\right], \ \gamma := \mathbb{E}\left[q(\|w_s\|G_1)G_1^2\right],$$ (43)

### A.5. Test Error Representation ("Scaling Laws")

We are now ready to state our main theoretical results, which is a generalization of Theorem 1.

**Remark 1.** *For simplicity of presentation, all our theoretical results only consider symmetric pruning strategies for which* $q(-t) \equiv q(t)$. *This includes the "keep hard" and "keep easy" pruning strategies considered in (Sorscher et al., 2022).*

**Proposition 2.** *Define the following quantities:*

$$m := \frac{m_0}{1 + \delta}, \quad m_0 := \frac{c^\top \bar{R}\Sigma w_0}{\|\Sigma^{1/2}w_0\|}$$ (44)

$$\nu := \frac{\nu_0}{(1 + \delta)^2}, \quad \nu_0 := \frac{p}{n} \operatorname{tr} \Sigma\Sigma' + c^\top \Sigma' c - \frac{2c^\top \bar{R}c}{1 + \delta} \frac{1}{n} \operatorname{tr} \Sigma\Sigma',$$ (45)

$$\text{with } c := \mathbb{E}[q_i y_i x_i] = \mathbb{E}_{(x,y) \sim P}[q(x^\top w_s)yx], \quad \Sigma' := \mathbb{E}[R\Sigma R].$$ (46)

*Then, in the limit* (14)*, the test error of* $\hat{w}$ *is given by*

$$E_{test}(\hat{w}) \to \frac{1}{\pi} \arccos\left(|m_0|/\sqrt{\nu_0}\right).$$ (47)

## B. Proof of Proposition 2

The proof follows standard (Couillet & Liao, 2022; Firdoussi et al., 2024) "leave-one-out" techniques which are now standard for analyses based on random matrix theory.

## B.1. Main Idea

For a random test point $(x, y) \sim P_*$, we can write

$$yx^\top \hat{w} = yz^\top \Sigma^{1/2} \hat{w} = sign(z^\top \Sigma^{1/2} w_0) z^\top \Sigma^{1/2} \hat{w}.$$

Write $\Sigma^{1/2}\hat{w} = \alpha\Sigma^{1/2}w_0 + r$, where $r = \Sigma^{1/2}\hat{w} - \alpha\Sigma^{1/2}w_0$ and $\alpha \geq 0$ is to be determined. Observe that $r$ is perpendicular to $\Sigma^{1/2}w_0$ iff $r^\top \Sigma^{1/2} w_0 = \hat{w}^\top \Sigma w_0 - \alpha \|\Sigma^{1/2}w_0\|^2 = 0$ iff

$$\alpha = \hat{w}^\top w_0 / \|\Sigma^{1/2}w_0\|^2. \tag{48}$$

With this choice of $\alpha$, one computes

$$yx^\top \hat{w} = \alpha yz^\top \Sigma^{1/2} w_0 + yz^\top r. \tag{49}$$

Because $r$ is perpendicular to $\Sigma^{1/2}w_0$, we know that the above is a sum of two independent random variables.

For the first summand in (49), observe that

$$yz^\top \Sigma^{1/2} w_0 = yx^\top w_0 = sign(x^\top w_0) x^\top w_0 = |x^\top w_0|,$$

which has the same distribution as $\|\Sigma^{1/2}w_0\||G_1|$ for $G_1 \sim N(0,1)$. For the second summand, it has the same distribution as distribution $\|r\|G_2$ where $G_2 \sim \mathcal{N}(0,1)$ and $\|r\|^2 = \|\Sigma^{1/2}\hat{w}\|^2 - \alpha^2\|\Sigma^{1/2}w_0\|^2$. It follows that if $\alpha > 0$,

$$
\begin{aligned}
E_{test}(\hat{w}) = \mathbb{P}_{x,y}(yx^\top \hat{w} \leq 0) &= \mathbb{P}(\alpha\|\Sigma^{1/2}w_0\||G_1| + \|r\|G_2 \leq 0) \\
&= \mathbb{P}(\alpha\|\Sigma^{1/2}w_0\||G_1| + \|r\|G_2 \leq 0, \ G_2 < 0) \\
&= \mathbb{P}(|G_2/G_1| \geq \eta, \ G_2 < 0) \text{ with } \eta := \alpha\|\Sigma^{1/2}w_0\|/\|r\| \\
&= \mathbb{P}(G_2 < 0)\mathbb{P}(|T| \geq \eta) \text{ with } T := G_2/G_1 \sim Cauchy(0,1) \\
&= \frac{1}{2} \cdot 2\mathbb{P}(T \geq \eta) = \mathbb{P}(T \geq \eta) \\
&= 1 - (\frac{1}{2} + \frac{1}{\pi}\arctan\eta) = \frac{1}{\pi}(\pi/2 - \arctan\eta) \\
&= \frac{1}{\pi}\arccos(\frac{\eta}{\sqrt{1+\eta^2}}) = \frac{1}{\pi}\arccos(\frac{\alpha\|\Sigma^{1/2}w_0\|}{\|\Sigma^{1/2}\hat{w}\|}).
\end{aligned}
$$

Similarly, if $\alpha < 0$, we get the same expression with $\alpha$ replaced by $-\alpha$. Therefore, irrespective of $\alpha$, we have

$$E_{test}(\hat{w}) = \frac{1}{\pi}\arccos(\frac{|\alpha|\|\Sigma^{1/2}w_0\|}{\|\Sigma^{1/2}\hat{w}\|}). \tag{50}$$

It remains to estimate the random quantities $|\alpha|$ and $\|\Sigma^{1/2}\hat{w}\|$, in the asymptotic limit (14).

## B.2. Leave-One-Out Arguments

We start with the Woodbury identity tells us that

$$
\begin{aligned}
Rx_i = (X^\top DX/n + \lambda I_d)^{-1}x_i &= (\sum_{j=1}^{n} q_j x_j x_j^\top /n + \lambda I_d)^{-1}x_i \\
&= (R_{-i}^{-1} + q_i x_i x_i^\top /n)^{-1}x_i = \frac{R_{-i}x_i}{1 + q_i x_i^\top R_{-i}x_i/n},
\end{aligned}
$$

where $R_{-i} := (n^{-1}\sum_{j\neq i} q_j x_j x_j^\top + \lambda I_d)^{-1}$ is a version of the resolvent matrix constructed without the $i$th data point. This "leave-one-out" trick is well-known in random matrix theory calculations.

On the other hand $q_i x_i^\top R_{-i}x_i/n$ concentrates around its mean which is

$$\mathbb{E}[q_i x_i^\top R_{-i}x_i/n] = \text{tr}\left(\mathbb{E}[q_i x_i x_i^\top]R_{-i}/n\right) = \frac{\alpha}{n}\text{tr}\,\Sigma R_{-i} \simeq \delta,$$

$$\text{with } \delta := \frac{p}{n}\text{tr}\,\Sigma\bar{R}, \quad p := \mathbb{E}[q_i].$$

Therefore, we have the following identities holding for every $i, j \in [n]$ with $i \neq j$:

$$Rx_i \simeq \frac{R_{-i}x_i}{1 + \delta}, \tag{51}$$

$$R_{-i} \simeq R_{-ij} - \frac{R_{-ij}x_jx_j^\top R_{-ij}}{1 + \delta}. \tag{52}$$

Now, let $x$ be a random test point from class $y$, independent of training data. For later use, note that

$$yx^\top \hat{w} = \frac{1}{n}\sum_{i=1}^n q_i y_i yx^\top Rx_i = \frac{1}{n}\sum_{i=1}^n q_i y_i yx^\top Rx_i$$

$$= \frac{1}{(1+\delta)n}\sum_{i=1}^n q_i y_i yx^\top R_{-i}x_i. \tag{53}$$

## B.3. Asymptotics of $\|\Sigma^{1/2}\hat{w}\|^2$

Note that $\|\Sigma^{1/2}\hat{w}\|^2 = \mathbb{E}_{x,y}[(yx^\top\hat{w})^2] = \mathbb{E}[(x^\top\hat{w})^2]$. Squaring (53) gives

$$(x^\top\hat{w})^2 = \frac{1}{(1+\delta)^2 n^2}\sum_{i=1}^n q_i \cdot (x^\top R_{-i}x_i)^2 + \frac{1}{(1+\delta)^2 n^2}\sum_{i \neq j} q_i q_j y_i y_j (x^\top R_{-i}x_i)(x^\top R_{-j}x_j)$$

For the expectation first some, note that

$$\frac{1}{n}\mathbb{E}\left[q_i \cdot (x^\top R_{-i}x_i)^2\right] = \frac{1}{n}\mathbb{E}[q_i x^\top R_{-i}x_i x_i^\top R_{-i}x] = \frac{1}{n}\operatorname{tr}\left(\mathbb{E}\left[xx^\top\right]\mathbb{E}\left[q_i R_{-i}x_i x_i^\top R_{-i}\right]\right) = \frac{p}{n}\operatorname{tr}\Sigma\Sigma',$$

with $\Sigma' := \mathbb{E}[R\Sigma R]$. We deduce that

$$\mathbb{E}\frac{1}{(1+\delta)^2 n^2}\sum_{i=1}^n q_i \cdot (x^\top R_{-i}x_i)^2 = \frac{1}{(1+\delta)^2}\frac{p}{n}\operatorname{tr}\Sigma\mathbb{E}[R\Sigma R]$$

$$= \frac{p}{(1+\delta)^2} \cdot \begin{cases} n^{-1}\operatorname{tr}\mathbb{E}[R^2]\Sigma, & \text{if isotropic,} \\ \text{hard life!,} & \text{otherwise.} \end{cases}$$

Now, let $i, j \in [n]$ with $i \neq j$. One computes

$$\mathbb{E}\left[q_i q_j y_i y_j \cdot (x^\top R_{-i}x_i)(x^\top R_{-j}x_j)\right] = \frac{1}{1+\delta}\mathbb{E}\left[q_i q_j y_i y_j x_i^\top T_{ij}\Sigma T_{ji}x_j\right],$$

$$= \frac{1}{1+\delta}(A_1 - A_2 - A_3 + A_4),$$

$$\text{where } T_{ij} := R_{-ij} - S_{ij}/n,$$

$$S_{ij} := \frac{R_{-ij}x_jx_j^\top R_{-ij}}{1+\delta},$$

$$A_1 := \mathbb{E}[q_i q_j y_i y_j x_i^\top R_{-ij}\Sigma R_{-ij}x_j],$$

$$A_2 := \frac{1}{(1+\delta)n}\mathbb{E}[q_i q_j y_i y_j x_i^\top S_{ij}\Sigma R_{-ij}x_j],$$

$$A_3 := \frac{1}{(1+\delta)n}\mathbb{E}[q_i q_j y_i y_j x_i^\top R_{-ij}\Sigma S_{ji}x_j],$$

$$A_4 := \frac{1}{(1+\delta)^2 n^2}\mathbb{E}[q_i q_j y_i y_j x_i^\top S_{ij}\Sigma S_{ji}x_j]$$

We now compute the terms $A_1, A_2, A_3, A_4$.

$$A_1 = \mathbb{E}\left[q_i q_j y_i y_j x_i^\top R_{-ij} \Sigma R_{-ij} x_j\right] = \mathbb{E}\left[q_i q_j y_i y_j x_i^\top R \Sigma R x_j\right]$$
$$= \operatorname{tr}\left(\mathbb{E}\left[(q_j y_j x_j)(q_i y_i x_i)^\top\right]\mathbb{E}\left[R\Sigma R\right]\right) = c^\top \Sigma' c,$$

where $\Sigma' := \mathbb{E}[R\Sigma R]$.

Similarly, $A_3 = A_2$ with

$$A_2 = \mathbb{E}\left[q_i q_j y_i y_j x_i^\top S_{ij} \Sigma R_{-ij} x_j\right] = \frac{1}{(1+\delta)n}\mathbb{E}\left[q_i q_j y_i y_j x_i^\top R_{-ij} x_j x_j^\top R_{-ij} \Sigma R_{-ij} x_j\right]$$
$$= \frac{1}{(1+\delta)n}\operatorname{tr}\left(\mathbb{E}\left[q_i q_j y_i y_j x_j x_i^\top R_{-ij} x_j x_j^\top\right]\mathbb{E}\left[R_{-ij}\Sigma R_{-ij}\right]\right)$$

Now, computes

$$\mathbb{E}\left[q_i y_i q_j y_j x_i^\top R_{-ij} x_j\right] = \mathbb{E}\left[(q_i y_i x_i)^\top R_{-ij}(q_j y_j x_j)\right] = c^\top \mathbb{E}\left[R_{-ij}\right]c \simeq c^\top \mathbb{E}\left[R\right]c \simeq c^\top \bar{R}c,$$
$$\mathbb{E}[R_{-ij}\Sigma R_{-ij}] \simeq \mathbb{E}[R\Sigma R] =: \Sigma',$$

from which it follows that

$$A_3 = A_2 \simeq \frac{c^\top \bar{R}c}{1+\delta}\frac{1}{n}\operatorname{tr}\Sigma\Sigma'. \tag{54}$$

Finally, it is easy to show that $A_4 = O(1/n) = o(1)$.

Putting things together, we deduce that

$$\mathbb{E}[\|\Sigma^{1/2}\hat{w}\|^2] \simeq \nu := \frac{\nu_0}{(1+\delta)^2}, \tag{55}$$

where $\nu_0 \geq 0$ is as Proposition 2.

### B.4. Asymptotics of $\alpha$

Proceeding as in the computation of the asymptotics of $\|\Sigma^{1/2}\hat{w}\|^2$ above, we can show that

$$\|\Sigma^{1/2}w_0\|^4 \mathbb{E}\alpha^2 = \mathbb{E}(\hat{w}^\top \Sigma w_0)^2 = \mathbb{E}\hat{w}^\top \Sigma w_0 w_0^\top \Sigma \hat{w} \simeq \frac{1}{(1+\delta)^2}c^\top R\Sigma w_0 w_0^\top \Sigma Rc \simeq \frac{(c^\top \bar{R}\Sigma w_0)^2}{(1+\delta)^2}.$$

On the other hand,

$$\|\Sigma^{1/2}w_0\|^2 \mathbb{E}\alpha = \mathbb{E}\hat{w}^\top \Sigma w_0 \simeq \frac{1}{1+\delta}\mathbb{E}\frac{1}{n}\sum_i q_i y_i x_i^\top R_{-i}\Sigma w_0$$
$$\simeq \frac{1}{1+\delta}\mathbb{E}[q_i y_i x_i^\top R_{-i}\Sigma w_0]$$
$$= \frac{1}{1+\delta}\mathbb{E}[q_i y_i x_i]^\top \mathbb{E}[R_{-i}]\Sigma w_0$$
$$\simeq \frac{c^\top \bar{R}\Sigma w_0}{1+\delta}.$$

Thus, the variance of $\alpha$ is vanishing, and we deduce that

$$\alpha \simeq \mathbb{E}\alpha \simeq \frac{c^\top \bar{R}\Sigma w_0}{(1+\delta)\|\Sigma^{1/2}w_0\|^2} =: \frac{m_0}{(1+\delta)\|\Sigma^{1/2}w_0\|}, \tag{56}$$

where $m_0$ is as given Proposition 2.

## B.5. Final Step (Proof of Proposition 2)

Combining (50), (55), and (56) completes the prove of Proposition 2. $\qquad\square$

## B.6. An Important Lemma

The following result computes the mean vectors $\mu$ and $c$.

**Lemma 4.** *Let $\rho \in [-1,1]$ be the cosine of the angle between $\bar{w}_s := \Sigma^{1/2} w_s$ and $\bar{w}_0 := \Sigma^{1/2} w_0$. Let $u$ be the unit-vector in the direction of $\bar{w}_s$ and let $v$ be its completion to an orthonormal basis for the span of $\bar{w}_s$ and $\bar{w}_0$ (if $\bar{w}_s$ and $\bar{w}_0$ are parallel, i.e if $\rho = \pm 1$, we simply set $v = 0$).*

$$\mu := \mathbb{E}_{(x,y)\sim P}[yx], \quad c := \mathbb{E}_{(x,y)\sim P}[q(x^\top w_s)yx] \tag{57}$$

*Then, $\mu = \sqrt{2/\pi} \cdot \Sigma w_0 / \|w_0\|_\Sigma$, and $c = \tilde{\beta} u + \beta v$, where*

$$\tilde{\beta} = \beta_1 := 2\mathbb{E}\left[q(\|\bar{w}_s\|G)\Phi(\tau G)G\right], \quad \beta = \beta_2 := 2\mathbb{E}\left[q(\|\bar{w}_s\|G)\varphi(\tau G)\right], \quad with\ G \sim \mathcal{N}(0,1). \tag{58}$$

*In particular, when $\rho = \pm 1$ (i.e pruning along the data generator),*

$$\beta_1 = \mathbb{E}[q(\|\bar{w}_s\|G)|G|], \quad \beta_2 = 0. \tag{59}$$

*Proof.* Observe that by instead considering $\Sigma^{-1/2}\mu$, $\Sigma^{-1/2}c$, and defining $v := \Sigma^{1/2} w_s$ and $u := \Sigma^{1/2} w_0$ when computing $\mu$, and then $u = \Sigma^{1/2} w_0$ when computing $c$, we reduce the problem to the isotropic case $x \sim \mathcal{N}(0, I_d)$.

So let $u = \Sigma^{1/2} w_0$, and WLOG, assume $u$ is aligned with the first canonical axis in $\mathbb{R}^d$, i.e $u = \|u\|e_1$. Write $x = (x_1, x_\perp)$ and $v = (v_1, v_\perp)$, where $x_\perp := \sum_{j=2}^d x_j e_j \in \mathbb{R}^{d-1}$, and $v_\perp := \sum_{j=2}^d v_j e_j \in \mathbb{R}^{d-1}$. It is clear that $x^\top u = \|u\|x_1$, and $x^\top v = v_1 x_1 + g$, where $g = x_\perp^\top v_\perp$. Furthermore, $x_1$ and $g$ are independent with distributions $\mathcal{N}(0,1)$ and $\mathcal{N}(0, \|v_\perp\|^2)$ respectively. It follows that

$$\Sigma^{-1/2}\mu = \mathbb{E}\left[sign(x^\top u)x\right] = \mathbb{E}\left[sign(\|u\|x_1)x_1\right]e_1 = \mathbb{E}\left[|x_1|\right]e_1$$
$$= \sqrt{\frac{2}{\pi}}e_1 = \sqrt{\frac{2}{\pi}}\frac{u}{\|u\|} = \sqrt{\frac{2}{\pi}}\frac{\Sigma^{1/2} w_0}{\|w_0\|_\Sigma},$$

from which we deduce the prescribed formula for the vector $\mu$. This proves the first part of the claim.

We now establish the formula $c = \beta_1 u + \beta_2 v$. The proof for the formula for $\mu$ follows a similar (but simpler) path.

Observe that by instead considering $\Sigma^{-1/2}c$, we reduce the problem to the isotropic case $x \sim \mathcal{N}(0, I_d)$. We can explicitly write

$$u = \frac{\bar{w}_s}{\|\bar{w}_s\|}, \quad v = \frac{\Pi^\perp \bar{w}_0}{\|\Pi^\perp \bar{w}_0\|}, \quad \rho = \frac{\bar{w}_s^\top \bar{w}_0}{\|\bar{w}_s\|\|\bar{w}_0\|}, \tag{60}$$

where $\Pi = uu^\top$ and $\Pi^\perp = I_d - \Pi$. One can decompose $x = G_1 u + G_2 v + G_\perp$ and $\bar{w}_0 = c_1 u + c_2 v + c_\perp$

$$G_1 := x^\top u, \quad G_2 := x^\top v, \quad G_\perp := P^\perp x, \tag{61}$$
$$c_1 := w_0^\top u, \quad c_2 := x^\top v, \quad c_\perp := P^\perp \Sigma^{1/2} w_0, \tag{62}$$

where $P$ is the projector onto the span of $u$ and $v$. Note that $G_1$, $G_2$, and $G_\perp$ forms a set of independent random variables. Moreover, $G_1$ and $G_2$ have distribution $\mathcal{N}(0,1)$, while $G_\perp$ has distribution $\mathcal{N}(0, I_{d-2})$. We obtain

$$\mathbb{E}[q(x^\top w_s)sign(x^\top w_0)x] = \mathbb{E}\left[q(x^\top w_s)sign(x^\top w_0)x\right] = \mathbb{E}\left[q(x^\top w_s)sign(x^\top w_0)x\right] \tag{63}$$
$$= \mathbb{E}\left[q(\|w_s\|G_1)sign(c_1 G_1 + c_2 G_2)G_1\right] \cdot u \tag{64}$$
$$+ \mathbb{E}\left[q(\|w_s\|G_1)sign(c_1 G_1 + c_2 G_2)G_2\right] \cdot v \tag{65}$$
$$+ \mathbb{E}\left[q(\|w_s\|G_1)sign(c_1 G_1 + c_2 G_2)G_\perp\right]. \tag{66}$$

Now, due independence, the third term decomposes as

$$\mathbb{E}\left[q(\|w_s\|_\Sigma \cdot G_1)sign(c_1 G_1 + c_2 G_2)\right] \cdot \mathbb{E}\left[G_\perp\right] = 0.$$

We deduce that

$$\mathbb{E}[q(x^\top w_s)sign(x^\top w_0)x] = \beta_1 u + \beta_2 v,$$

where $\beta_1$ and $\beta_2$ are as specified in the lemma and we have used the fact that

$$c_1/\|\bar{w}_0\| = \rho, \quad c_2/\|\bar{w}_0\| = \sqrt{1-\rho^2}.$$

In particular, if $\rho = \pm 1$ (meaning that $w_0$ and $w_s$ are parallel), then

$$\beta_k = \mathbb{E}\left[sign(\pm G_1)q(\|\bar{w}_s\| \cdot G_1)G_k\right] = \begin{cases} \pm\beta, & \text{if } k = 1, \\ 0, & \text{otherwise.} \end{cases} \tag{67}$$

We now compute the coefficients $\beta_1$ and $\beta_2$. Observe that thanks to Lemma 5, one has

$$\mathbb{E}[sign(G_3) \mid G_1] = \mathbb{E}[sign(\rho G_1 + \sqrt{1-\rho^2}G_2) \mid G_1] = 2\Phi\left(\tau G_1\right) - 1,$$
$$\mathbb{E}[sign(G_3)G_2) \mid G_1] = \mathbb{E}[sign(\rho G_1 + \sqrt{1-\rho^2}G_2)G_2 \mid G_1] = 2\varphi(\tau G_1).$$

Therefore, with $r := \|\bar{w}_s\|$, we have

$$\beta_1 := \mathbb{E}[q(rG_1)sign(G_3)G_1] = 2\mathbb{E}\left[q(rG_1)\Phi\left(\tau G_1\right)G_1\right] - \mathbb{E}\left[q(rG_1)G_1\right] = 2\mathbb{E}\left[q(rG_1)\Phi\left(\tau G_1\right)G_1\right],$$
$$\beta_2 := \mathbb{E}[q(rG_1)sign(G_3)G_2] = 2\mathbb{E}\left[q(rG_1)\varphi(\tau G_1)\right],$$

where we have used the oddness of the function $t \mapsto tq(rt)$ in the last equation on the first line. □

**Lemma 5.** *Let $G \sim \mathcal{N}(0, 1)$, and let $a, b \in \mathbb{R}$ with $a \neq 0$. Then,*

$$\mathbb{E}[sign(aG + b)] = 2\Phi(b/|a|) - 1, \quad \mathbb{E}[sign(aG + b)G] = 2\varphi(b/a). \tag{68}$$

*Furthermore, it holds that*

$$\lim_{a \to 0} \mathbb{E}[sign(aG + b)] = sign(b), \quad \lim_{a \to 0} \mathbb{E}[sign(aG + b)G] = 0. \tag{69}$$

*Proof.* Indeed, one computes

$$\mathbb{E}[sign(aG + b)] = \mathbb{P}(aG + b > 0) - \mathbb{P}(aG + b < 0) = 2\mathbb{P}(aG > -b) - 1$$
$$= \begin{cases} 2\mathbb{P}(G > -b/a) - 1 = 2\Phi(b/a) - 1, & \text{if } a > 0, \\ 2\mathbb{P}(G < -b/a) - 1 = 2\Phi(-b/a) - 1, & \text{if } a < 0. \end{cases}$$

We deduce that $\mathbb{E}[sign(aG + b)] = 2\Phi(b/|a|) - 1$ as claimed. □

## C. Proof of Lemma 1 and Lemma 2

**"Keep Hard" Examples (Lemma 1).** Let $b = \tau$, $t = \sqrt{1 + b^2} = \sqrt{1 + \tau^2} = 1/\sqrt{1-\rho^2}$. Using Lemma 4 and standard formulae[1] for the anti-derivative of the function $z \mapsto z\varphi(bz)\varphi(z)$

$$\beta = \beta_2 = 2\mathbb{E}\left[q(rG)\varphi(\tau G)\right] = 2\int_{-\alpha}^{\alpha} \varphi(\tau z)\varphi(z)\mathrm{d}z = \frac{2}{t}\varphi(0)\Phi(tz)\Big]_{z=-\alpha}^{\alpha}$$
$$= 2\sqrt{1-\rho^2}\varphi(0)\left(2\Phi(\alpha/\sqrt{1-\rho^2}) - 1\right) = 2\varphi(0)\sqrt{1-\rho^2}\epsilon_2.$$

---

[1]For example, see Wikipedia `https://en.wikipedia.org/wiki/List_of_integrals_of_Gaussian_functions`.

On the other hand, we have $\tilde{\beta} = \beta_1 = 2\mathbb{E}\left[q(rG)\Phi(\tau G)G\right] = 2\int_{-\alpha}^{\alpha} z\Phi(\tau z)\varphi(z)\mathrm{d}z$ with

$$
\int_{-\alpha}^{\alpha} z\Phi(\tau z)\varphi(z)\mathrm{d}z = (b/t)\varphi(0)\Phi(tz) - \varphi(z)\Phi(bz)\Big]_{z=-\alpha}^{\alpha}
$$

$$
= (b/t)\varphi(0)(2\Phi(t\alpha) - 1) - \varphi(\alpha)(2\Phi(b\alpha) - 1)
$$

$$
= \rho\varphi(0)(2\Phi(\alpha/\sqrt{1-\rho^2}) - 1) - \varphi(\alpha)(2\Phi(\tau\alpha) - 1)
$$

$$
= \rho\varphi(0)\epsilon_1 - \varphi(\alpha)\epsilon_2,
$$

which proves Lemma 1

**"Keep Easy" Examples (Lemma 2).** Indeed, since $q_{KE} = 1 - q_{KH}$, we know from the previous lemma (KH strategy) that

$$
\tilde{\beta}(q_{KE}) = 2\mathbb{E}\left[q_{KE}(rG)\Phi(\tau G)G\right] = 2\mathbb{E}\left[\Phi(\tau G)G\right] - 2\mathbb{E}\left[q_{KH}(rG)\Phi(\tau G)G\right]
$$

$$
= 2\mathbb{E}\left[\Phi(\tau G)G\right] - 2\tilde{\beta}(q_{KH}) = 2(\rho\varphi(0) - \varphi(\alpha)) - \tilde{\beta}(q_{KH})
$$

$$
= 2\rho\varphi(0) - 2\rho\varphi(0)\epsilon_1(q_{KH}) + 2\varphi(\alpha)\epsilon_2(q_{KH})
$$

$$
= 2(\rho\varphi(0)(1 - \epsilon_1(q_{KH})) + \varphi(\alpha)\epsilon_2(q_{KH}))
$$

$$
= 2(\rho\varphi(0)\epsilon_1 + \varphi(\alpha)\epsilon_2).
$$

The computation of $\beta_2(q_{KE})$ uses a completely analogous idea:

$$
\beta(q_{KE}) = 2\mathbb{E}[q_{KE}(rG)\varphi(\tau G)] = 2\mathbb{E}[\varphi(\tau G)] - 2\mathbb{E}[q_{KH}(rG)\varphi(\tau G)]
$$

$$
= 2\varphi(0)\sqrt{1-\rho^2} - 2\beta(q_{KH})
$$

$$
= 2\left(\varphi(0)\sqrt{1-\rho^2} - \varphi(0)\sqrt{1-\rho^2}\epsilon_1(q_{KH})\right)
$$

$$
= 2\varphi(0)\sqrt{1-\rho^2}\,(1 - \epsilon_1(q_{KH}))
$$

$$
= 2\varphi(0)\sqrt{1-\rho^2}\epsilon_1(q_{KE})
$$

This proves Lemma 2. $\qquad\square$

### C.1. Proof of Proposition 1

Using Theorem 4 of Liao and Mahoney's "Hessian Eigenspectra of More Realistic Nonlinear Models" https://arxiv.org/abs/2103.01519 and some basic manipulations, we can write

$$
R \simeq \bar{R}, \tag{70}
$$

$$
\text{where } \bar{R}^{-1} = \mathbb{E}_x\left[\frac{q}{1 + q\delta(z)}(\Sigma^{1/2}\Pi_\perp\Sigma^{1/2} + \alpha\alpha^\top)\right] - zI_d, \tag{71}
$$

where $q := q(x^\top w_s)$ for $x \sim \mathcal{N}(0, \Sigma)$, $\alpha := \Sigma^{1/2}\Pi x$. Since $q$ is Bernoulli with mean $p := \mathbb{P}(q = 1)$, it is clear that

$$
\mathbb{E}_x\left[\frac{q}{1 + q\delta(z)}\right] = \frac{p}{1 + \delta(z)} := t(z).
$$

This further gives

$$
\bar{R}^{-1} = t(z)\Sigma^{1/2}\Pi_\perp\Sigma^{1/2} - zI_d + \Sigma^{1/2}\Pi K\Pi\Sigma^{1/2},
$$

$$
\text{with } K := \mathbb{E}_u\left[\frac{q(u^\top v)}{1 + q(u^\top v)\delta(z)}uu^\top\right], \tag{72}
$$

where $u := \Sigma^{-1/2}x \sim \mathcal{N}(0, I_d)$ and $v := \Sigma^{1/2}w_s$.

Now, to determine the matrix $K$, we first rewrite $u = (u_{/\!/}, u_\perp)$ and $v = (v_1, v_\perp)$, where

$$
u_{/\!/} := \frac{u^\top w_s}{\|w_s\|} \in \mathbb{R}, \quad v_1 := \frac{v^\top w_s}{\|w_s\|} \in \mathbb{R}, \tag{73}
$$

$$
u_\perp := \Pi_\perp u \in \mathbb{R}^{d-1}, \quad v_\perp := \Pi_\perp v \in \mathbb{R}^{d-1}. \tag{74}
$$

The advantage of this representation is that

- $u_\perp$ and $v_\perp$ are orthogonal to $w_s$.

- $u_\parallel$ and $u_\perp$ are statistically independent.

- $u_\parallel$ has distribution $\mathcal{N}(0, 1)$.

- $u_\perp$ has distribution $\mathcal{N}(0, I_{d-1})$.

By symmetry of the situation, we know that

$$K = s(z)\Pi + s_\perp(z)\Pi_\perp,$$
$$\text{where } s(z) := \mathbb{E}[h(w^\top g)G_1^2], \quad s_\perp(z) := \mathbb{E}[h(w^\top g)G_\perp^2]$$
$$w := (v_1, \|v_\perp\|) \in \mathbb{R}^2, \quad g := (G_1, G_\perp) \sim \mathcal{N}(0, I_2),$$
$$h(q) := \frac{q}{1 + q\delta(z)}.$$

Combining with (72), we get

$$\bar{R}^{-1} = \Sigma^{1/2}(a(z)I_d + b(z)\Pi)\Sigma^{1/2}, \tag{75}$$
$$\text{where } a(z) = t(z) - z, \quad t(z) = \frac{p}{1 + \delta(z)}, \quad b(z) = s(z) - t(z). \tag{76}$$

Now, using the *Matrix-Inversion Lemma*, one can obtain $\bar{R}$ from $\bar{R}^{-1}$ as follows:

$$\Sigma^{1/2}\bar{R}\Sigma^{1/2} = (a(z)I_d + b(z)\Pi)^{-1} = \frac{1}{a(z)}\left(I_d - \frac{b(z)/a(z)}{b(z)/a(z) + 1}\Pi\right) = \frac{1}{a(z)}\Pi_\perp + \frac{1}{b(z) + a(z)}\Pi.$$

It suffices to notice that $1/(b(z) + a(z)) = 1/(s(z) - z) = \tilde{m}(z)$ and $1/a(z) = \bar{m}(z)$ by definition, and the result follows.

### C.2. Proof of Theorem 1

Set $z = -\lambda$. Recall from Lemma 4 $c = \beta_1 u + \beta_2 v$. In Theorem 1, we have the identification $\beta = \beta_2$ and $\tilde{\beta} = \beta_1$. We know that $R \simeq \bar{R} = m(z)\Pi^\perp + \tilde{m}(z)\Pi$, where $\Pi = uu^\top$. One computes

$$m_0 = (w_0/\|w_0\|)^\top \bar{R}c = \frac{1}{\|w_0\|}w_0^\top \left(m(z)\Pi^\perp + \tilde{m}(z)\Pi\right)(\beta_1 u + \beta_2 v),$$

$$= \frac{1}{\|w_0\|}w_0^\top \left(\beta_1 \tilde{m}(z)u + \beta_2 m(z)v\right),$$

$$\text{with } \frac{w_0^\top u}{\|w_0\|} = \rho, \quad \frac{w_0^\top v}{\|w_0\|} = \frac{w_0^\top w_0/\|w_0\| - \rho\|w_0\|(u^\top w_0/\|w_0\|)}{\|w_0\|\sqrt{1 - \rho^2}}$$

$$= \frac{\rho - \rho^2}{\sqrt{1 - \rho^2}} = \sqrt{1 - \rho^2} =: \omega/\beta_2.$$

Putting things together gives $m_0 \simeq \omega m(z) + \tilde{\omega}\tilde{m}(z)$ as claimed.

Likewise, one computes

$$\frac{1}{n}\operatorname{tr} R^2 \simeq \frac{1}{n}\operatorname{tr}\left(m'(z)\Pi^\perp + \tilde{m}'(z)\Pi\right) \simeq \phi m'(z),$$
$$c^\top \bar{R}c = c^\top \left(m(z)\Pi^\perp + \tilde{m}(z)\Pi\right)c = (\beta_1 u + \beta_2 v)^\top (\tilde{m}(z)\Pi + m(z)\Pi^\perp)(\beta_1 u + \beta_2 v)$$
$$= \beta_2^2 m(z) + \beta_1^2 \tilde{m}(z) = \beta^2 m(z) + \tilde{\beta}^2 \tilde{m}(z) =: r(z),$$
$$c^\top \Sigma' c = c^\top \mathbb{E}[R^2]c \simeq c^\top \left(m'(z)\Pi^\perp + \tilde{m}'(z)\Pi\right)c = \beta^2 m'(z) + \tilde{\beta}^2 \tilde{m}'(z) = r'(z),$$

from which the claimed formula for $\nu$ follows. $\qquad\square$

### C.3. Proof of Corollary 1

As usual, set $z := -\lambda < 0$.

(A) For $\phi < p$, it is easy to see from formula (16) and Lemma 6 that in the limit $z \to 0$, one has

$$m(z) \to \frac{1}{p - \phi},$$
$$\bar{m}(z) \to 0,$$
$$\tilde{m}(z) \to \frac{p/\gamma}{p - \phi},$$
$$m'(z) \to \frac{p}{(p - \phi)^3},$$
$$\bar{m}'(z) \to \frac{1}{p - \phi},$$
$$\tilde{m}'(z) \to \frac{p/\gamma^2}{(p - \phi)^3} \left( p(p - \phi) + \phi\gamma \right) = \frac{p}{(p - \phi)^3} \left( (p - \phi)p/\gamma^2 + \phi/\gamma \right),$$
$$\frac{m'(z)}{1 + \phi m(z)} \to \frac{1}{(p - \phi)^2}.$$

Furthermore, with $m_0$ and $\nu_0$ as defined in Theorem 1, one computes

$$r(z) = \beta^2 m(z) + \tilde{\beta}^2 \tilde{m}(z) \to \beta^2 \frac{1}{p - \phi} + \tilde{\beta}^2 \frac{p/\gamma}{p - \phi} = \frac{r_0}{p - \phi},$$
$$r'(z) = \beta^2 m'(z) + \tilde{\beta}^2 \tilde{m}'(z) \to \beta^2 \cdot \frac{p}{(p - \phi)^3} + \tilde{\beta}^2 \cdot \frac{p/\gamma^2}{(p - \phi)^3} (p(p - \phi) + \phi\gamma) = \frac{r_0'}{(p - \phi)^3},$$

where $r_0$ and $r_0'$ are as defined in the claim. We deduce that $m_0 / \sqrt{\nu_0 - m_0^2} = a / \sqrt{b - a^2}$ and the result follows from Theorem 1.

(B) Now consider the case $\phi > p$. Observe that $m_0 = \sqrt{\nu_0 - m_0^2} = -zm_0 / \sqrt{z^2 - z^2 m_0^2}$. On the other hand, from (16) we know that

$$-zm(z) = \frac{\sqrt{(p - \phi - z)^2 - 4\phi z} - (p - \phi - z)}{2\phi} \tag{77}$$

Combining with Lemma 6, we deduce the following limits

$$-zm(z), z^2 m'(z) \to c_0 := 1 - p/\phi > 0,$$
$$\bar{m}'(z) \to \frac{p/\phi}{\phi - p},$$
$$-z\tilde{m}(z), z^2 \tilde{m}'(z) \to \frac{c_0}{\gamma/\phi + c_0},$$
$$\frac{-zm'(z)}{1 + \phi m(z)} \to \frac{1}{\phi}.$$

Furthermore, one computes

$$-zr(z) = \beta_2^2 \cdot (-zm(z)) + \beta_1^2 \cdot (-z\tilde{m}(z)) = \beta_2^2 c_0 + \beta_1^2 \frac{c_0}{\gamma/\phi + c_0} =: c_0 r_0,$$
$$z^2 r'(z) = \beta_2^2 z^2 m'(z) + \beta_1^2 z^2 \tilde{m}(z) = \beta_2^2 c_0 + \beta_1^2 \frac{c_0}{\gamma/\phi + c_0} = c_0 r_0,$$
$$-zm_0 = \sqrt{2/\pi} \cdot (-zm(z)\omega - z\tilde{m}(z)\tilde{\omega}) \to \sqrt{2/\pi} c_0 \cdot (\omega + \tilde{\omega}/(\gamma/\phi + c_0)) := a,$$
$$z^2 \nu_0 = p\phi z^2 m'(z) + z^2 r'(z) - 2\phi \frac{-zm'(z)}{1 + \phi m(z)} \cdot (-zr(z))$$
$$\to p\phi c_0 + r_0 c_0 - 2r_0 c_0 = c_0 \cdot (p\phi - r_0) =: b.$$

We deduce that

$$m_0/\sqrt{\nu_0 - m_0^2} = -zm_0/\sqrt{z^2\nu_0 - z^2 m_0^2} = a/\sqrt{b - a^2},$$

and the result follows from Theorem 1. □

**Lemma 6.** *We have the following identities:*

$$m'(z) = \frac{m(z)^2}{1 - (1 + \bar{m}(z))^2 \phi/p},$$

$$\bar{m}'(z) = \frac{p}{(z + \phi\bar{m}(z))^2/\bar{m}(z)^2 - p\phi} = \frac{p}{(\phi + 1/m(z))^2 - p\phi},$$

$$\tilde{m}'(z) = \tilde{m}(z)^2 \left( \frac{\gamma\phi m'(z)}{(1 + \phi m(z))^2} + 1 \right),$$

$$r'(z) = \beta^2 m'(z) + \tilde{\beta}^2 \tilde{m}'(z).$$

# D. Additional Experimental

## D.1. Choice of Generative Model

We evaluate the capabilities of four open-source large-scale pre-trained text-to-image models (Rombach et al., 2022) in a controlled setup to determine which one performs best for the image-classification task. Each synthetic image is generated with a simple prompt (class name). We create a dataset of size 130,000 examples and train a ViT-B model on the synthetic data. Our results (Table 2) show that LDM-1.5 outperforms its more recent counterparts, LDM-XL and LDM-2.1, despite being an older model. We hypothesize that this is due to the lower diversity of generations in more recent models. This finding is consistent with previous work (Astolfi et al., 2024), which observed lower diversity in more recent latent diffusion models. For all of our experiments, we use LDM-1.5 as it is the best performing model.

Table 2: Study on the choice of generative model for the task of ImageNet-100 classification with synthetic data. All experiments are trained for 50k iterations and the dataset size is a static size of 130k.

| Syn. Data Source | Real Val. Acc. |
|---|---|
| LDM-1.4 | 59.06 |
| LDM-1.5 | **59.24** |
| LDM-2.1 | 55.92 |
| LDM-XL | 52.8 |

## D.2. Ablations

$\omega = 0$ **vs** $\omega > 0$   To understand the effect of different components of our framework, we ablate the case where data is generated through the DP framework, but with a coefficient of zero for the term $\omega$. We also report results using different values of $\omega$. See results in Table 3 comparing row 1 with rows 2, 3 and 4.

**Incremental patience**   In our experiments, setting the maximum patience value ($T_{\max}$) to a fixed number resulted in the model requesting too much data when the size of the dataset was grown too big. For example, with a fixed patience of $T_{\max} = 7$, for an experiment with initial dataset of size 130k samples, monitoring the validation accuracy every 130k iterations, meant that in the beginning every example was seen on average 7 times before the patience reached $T_{\max}$. But as we generate more examples throughout the training, with a fixed patience value, each example would not be able to be seen even at least once. When the dataset grows to be 1.3 million, each example is seen on average of 0.7 times. This resulted in the model hitting the maximum patience very often. As a result, we incrementally increase the maximum patience value as the dataset increases in size. See Table 3 for the result that compares the two scenarios (comparing rows 1 and row 5). Note we found using an incremental patience to be significantly easier to tune. We often start with a patience of 1 and continue training. However, fixed patience requires more tuning depending on the size of the dataset and the number of training iterations.

**Dataset sampling probabilities**   One can assume that newly generated examples could be more valuable then previously generated examples. As a result, we experiment the case where every newly generated example has twice as much probability

Table 3: Ablation study on ImageNet-100. Given a baseline method with DP, we modify each component of the framework one-by-one and study the effect of each change. All the experiments are trained for 50k iterations and have the same final size.

| # | $\omega$ | $T_{max}$ | Sampling | Real Val. Acc. | Real tr. Acc. |
|---|---|---|---|---|---|
| 1 | 0.05 | inc. | uniform | 68.04 | 69.25 |
| 2 | **0** | inc. | uniform | 61.58 | 63.09 |
| 3 | **0.03** | inc. | uniform | 66.70 | 68.00 |
| 4 | **0.07** | inc. | uniform | 66.88 | 68.66 |
| 5 | 0.05 | **fixed** | uniform | 67.22 | 70.38 |
| 6 | 0.05 | inc. | **non-uni.** | 68.01 | 69.11 |

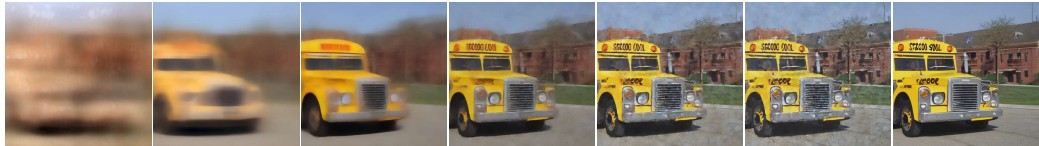

Figure 8: **Intermediate stages of reverse sampling.** Samples of the $x_0$ approximation using the DDIM sampler. While blurry, these intermediate samples provide sufficient gradients for entropy guidance, with key features like color and shape discernible even in early stages.

to be selected when sampling the data batch for a given iteration. We observe that having higher probability does not lead to statistically significant improvements. See results in Table 3 comparing rows 1 and 7.

### D.3. Intermediate stages of reverse sampling

In section 2 we mentioned that DDIM's $x_0$ approximation is a good approximation to guide the sampling process. In this section we plot these intermediate examples which are fed to the classifier to compute the entropy of the sample and use it for guidance in the sampling process. Figure 8 shows that although intermediate samples are noisy, they contain the key features.

### D.4. Studying the effect of $\omega$

In this section we study the effect of dynamically generating the data without entropy guidance versus generating it uniformly from the beginning. In the first scenario, we use the DP framework, with monitoring the patience variable but using an $\omega = 0$ which effectively generates with naive sampling. In the second case all the data is generated in advance and no data is added during training. As it can be see in Figure 9, there is close to no difference between generating all the data in advance or generating it dynamically if we allow for enough iterations for training.

In this experiment, we evaluate on ImageNet-100 validation set and train all the models for 50,000 iterations.

### D.5. Experimental Details

#### D.5.1. SCALING PLOTS

We have used the Warmup-Stable-Decay (WSD) learning rate scheduler (Hu et al., 2024), which stabilizes the learning rate throughout most of the training, ensuring effective adaptation to newly generated data. For ImageNet-100, we train on 4 nodes, each with 8 GPUs with a batchsize of 64. For ImageNet-1k, we train on 4 nodes, each with 8 GPUs with a batchsize of 128. For all the experiments, initial 10% of the iterations is done with linear-warmup and the last 20% of the iterations is for cool-down with Cosine Annealing. The intermediate steps are constant learning rate. For all these experiments we use $\lambda = 3$ and $\omega = 0.05$.

For ImageNet 100, the learning rate is 0.003 with an EMA momentum of 0.001. For ImageNet-1k, the learning rate is set to 0.0016 with an EMA momentum of 0.001. We also use label smoothing with a value of 0.11. We use Mixup with an alpha

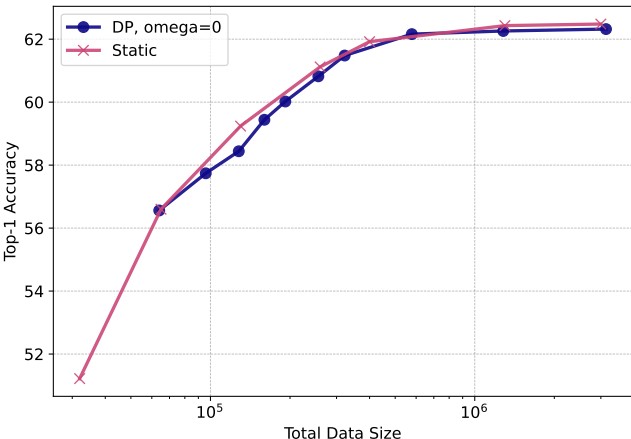

Figure 9: there is close to no difference between generating all the data in advance or generating it dynamically if we allow for enough iterations for training. Both cases have the same scaling behavior.

of 0.5 and CutMix with an alpha of 1.0. Furthermore, we use the AdamW optimizer.

Furthermore, for each setup in our experiments, we apply branch-outs. A branch-out is the same experiment as an initial setup except that it does not allow additional data starting from a specific epoch. The epoch is selected based on the times that the $T_{max}$ was hit. Meaning a branch out is just before additional data is added to the training set.

| N | P | k | N + kP | $\omega$ | Init. $T_{max}$ | Branch out Epoch | IN Val. | IN-Sk | IN-R* |
|---|---|---|---|---|---|---|---|---|---|
| 32000 | 16000 | 3 | 80000 | 0.05 | 6 | 662 | 59.48 | 31.49 | 58.92 |
| 32000 | 16000 | 4 | 96000 | 0.05 | 6 | 701 | 60.54 | 33.69 | 59.97 |
| 32000 | 16000 | 5 | 112000 | 0.05 | 6 | 767 | 61.80 | 35.03 | 61.24 |
| 32000 | 16000 | 6 | 128000 | 0.05 | 6 | 859 | 62.68 | 35.95 | 62.55 |
| 32000 | 16000 | 8 | 160000 | 0.05 | 6 | 951 | 64.40 | 38.08 | 63.87 |
| 64000 | 32000 | 6 | 256000 | 0.05 | 4 | 469 | 65.52 | 43.42 | 67.32 |
| 64000 | 32000 | 8 | 320000 | 0.05 | 4 | 606 | 66.28 | 44.33 | 67.94 |
| 64000 | 32000 | 11 | 416000 | 0.05 | 4 | 782 | 66.92 | 44.99 | 68.81 |
| 64000 | 32000 | 18 | 640000 | 0.05 | 4 | 1001 | 67.80 | 45.25 | 68.46 |
| 130000 | 130000 | 6 | 910000 | 0.05 | 14 | - | 68.28 | 45.06 | 70.87 |
| 130000 | 64000 | 27 | 1794000 | 0.05 | 5 | 494 | 68.46 | 46.33 | 71.04 |
| 130000 | 64000 | 47 | 3138000 | 0.05 | 5 | 618 | 68.88 | 45.76 | 71.26 |
| 64000 | 0 | - | 64000 | 0 | inf | - | 56.56 | 27.86 | 52.97 |
| 130000 | 0 | - | 130000 | 0 | inf | - | 59.44 | 33.32 | 55.95 |
| 260000 | 0 | - | 260000 | 0 | inf | - | 60.02 | 33.79 | 56.74 |
| 400000 | 0 | - | 400000 | 0 | inf | - | 61.92 | 36.03 | 59.75 |
| 2000000 | 0 | - | 2000000 | 0 | inf | - | 62.16 | 34.97 | 60.15 |
| 4000000 | 0 | - | 4000000 | 0 | inf | - | 62.32 | 36.43 | 60.89 |

Table 4: Details of the results reported in Figure 4 for the ImageNet-100 dataset. All the experiments are trained for 50k iterations. The variables are based on the notations defined in Algorithm 1. Note that $T_{max}$ is incremental.

## D.6. Visual examples

Below we provide additional examples of generations throughout time with different $\omega$ coefficients (x-axis) of [0.0001, 0.1, 0.3, 0.5, 0.7]. All samples are generated with the same seed. As from top to bottom the epoch number increases.

| N | P | k | N + kP | $\omega$ | Init. $T_{max}$ | Branch out Epoch | IN Val. | IN-Sk | IN-R |
|---|---|---|---|---|---|---|---|---|---|
| 160000 | 160000 | 1 | 320000 | 0.05 | 1 | 134 | 42.572 | 39.363 | 20.987 |
| 320000 | 160000 | 1 | 480000 | 0.05 | 1 | 191 | 44.880 | 41.987 | 23.095 |
| 320000 | 320000 | 1 | 640000 | 0.05 | 1 | 71 | 47.910 | 46.887 | 27.568 |
| 654000 | 654000 | 1 | 1308000 | 0.05 | 1 | 124 | 50.226 | 49.867 | 29.843 |
| 654000 | 654000 | 2 | 1962000 | 0.05 | 1 | 156 | 50.670 | 51.027 | 29.944 |
| 1300000 | 650000 | 10 | 7800000 | 0.05 | 1 | 246 | 50.908 | 49.820 | 31.217 |
| 654000 | 654000 | 19 | 13080000 | 0.05 | 1 | - | 51.198 | - | 16.776 |
| 320000 | 0 | - | 320000 | 0.0 | inf | - | 39.334 | 32.653 | 18.495 |
| 654000 | 0 | - | 654000 | 0.0 | inf | - | 42.514 | 33.883 | 21.303 |
| 1300000 | 0 | - | 1300000 | 0.0 | inf | - | 44.116 | 37.337 | 23.653 |
| 2600000 | 0 | - | 2600000 | 0.0 | inf | - | 45.006 | 38.667 | 24.298 |
| 10000000 | 0 | - | 10000000 | 0.0 | inf | - | 45.614 | 40.050 | 24.762 |
| 13000000 | 0 | - | 13000000 | 0.0 | inf | - | 45.628 | 40.357 | - |

Table 5: Details of the results reported in Figure 4 for the ImageNet-1k dataset. All the experiments are trained for 100k iterations. The variables are based on the notations defined in Algorithm 1. Note that $T_{max}$ is incremental.

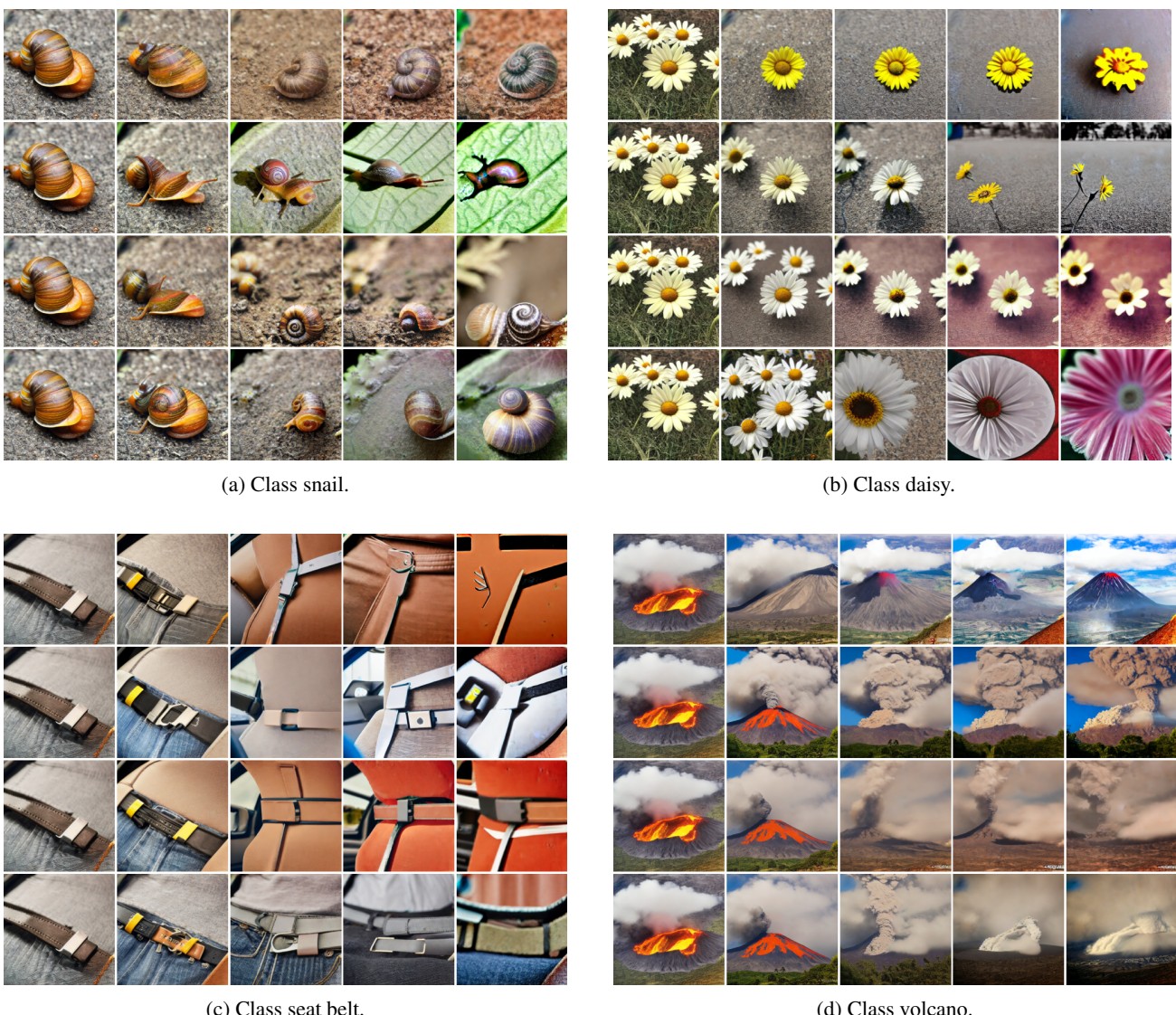

(a) Class snail.

(b) Class daisy.

(c) Class seat belt.

(d) Class volcano.

Figure 10: Examples of generated samples for different class prompts across training epochs, with varying entropy guidance coefficient ($\omega$) (left to right) as the training progresses (top to bottom).

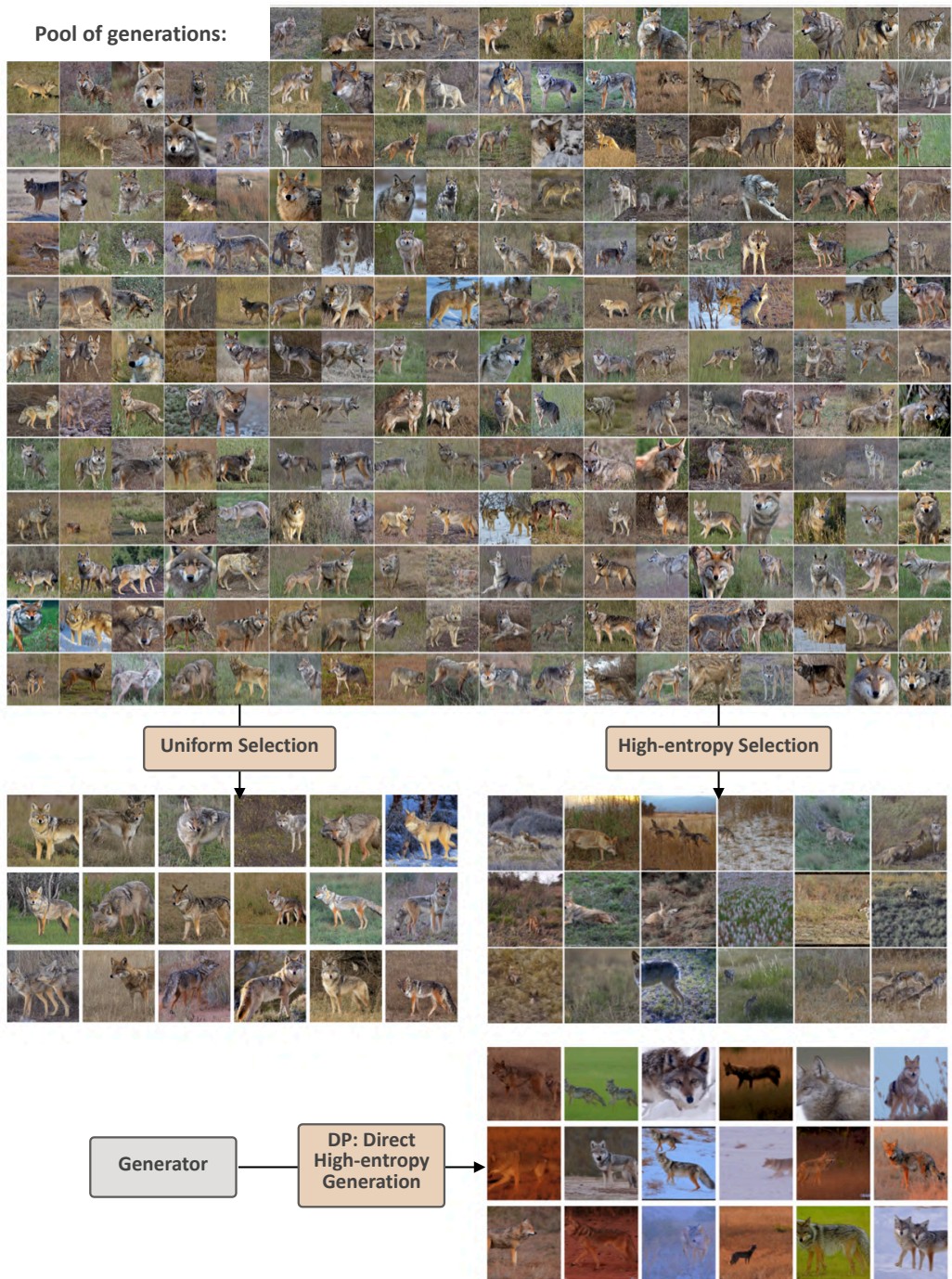

Figure 11: **Efficient and Diverse Sampling with DP:** Instead of inefficiently over-sampling and selecting high-entropy examples, DP directly generates high-entropy samples. This not only improves computational efficiency but also results in greater visual diversity.

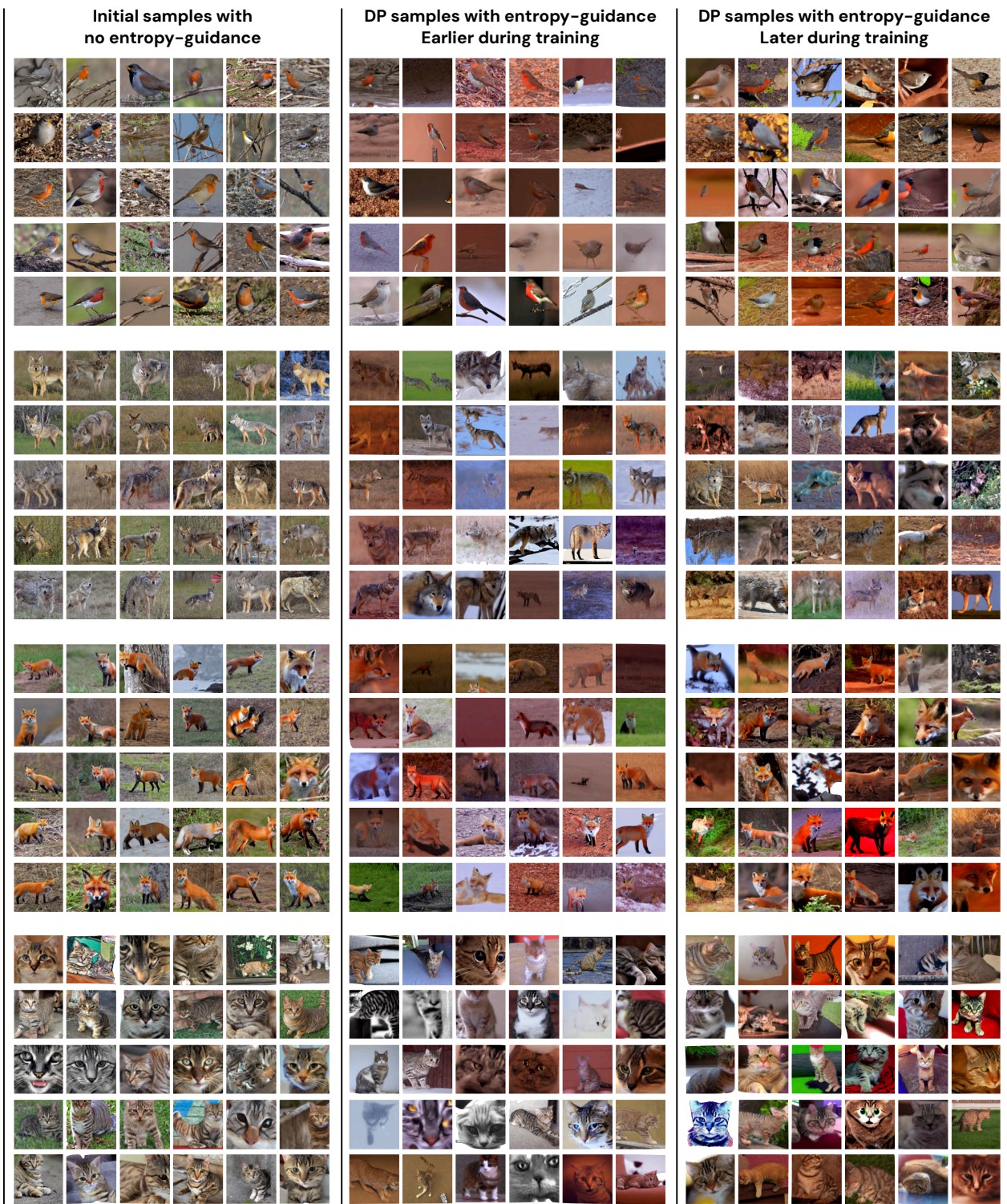

Figure 12: **Evolution of High-Entropy Samples During Training:** Early-stage generations show mainly color diversity, while later stages exhibit a richer set of transformations, aligning with the classifier's evolving uncertainties.

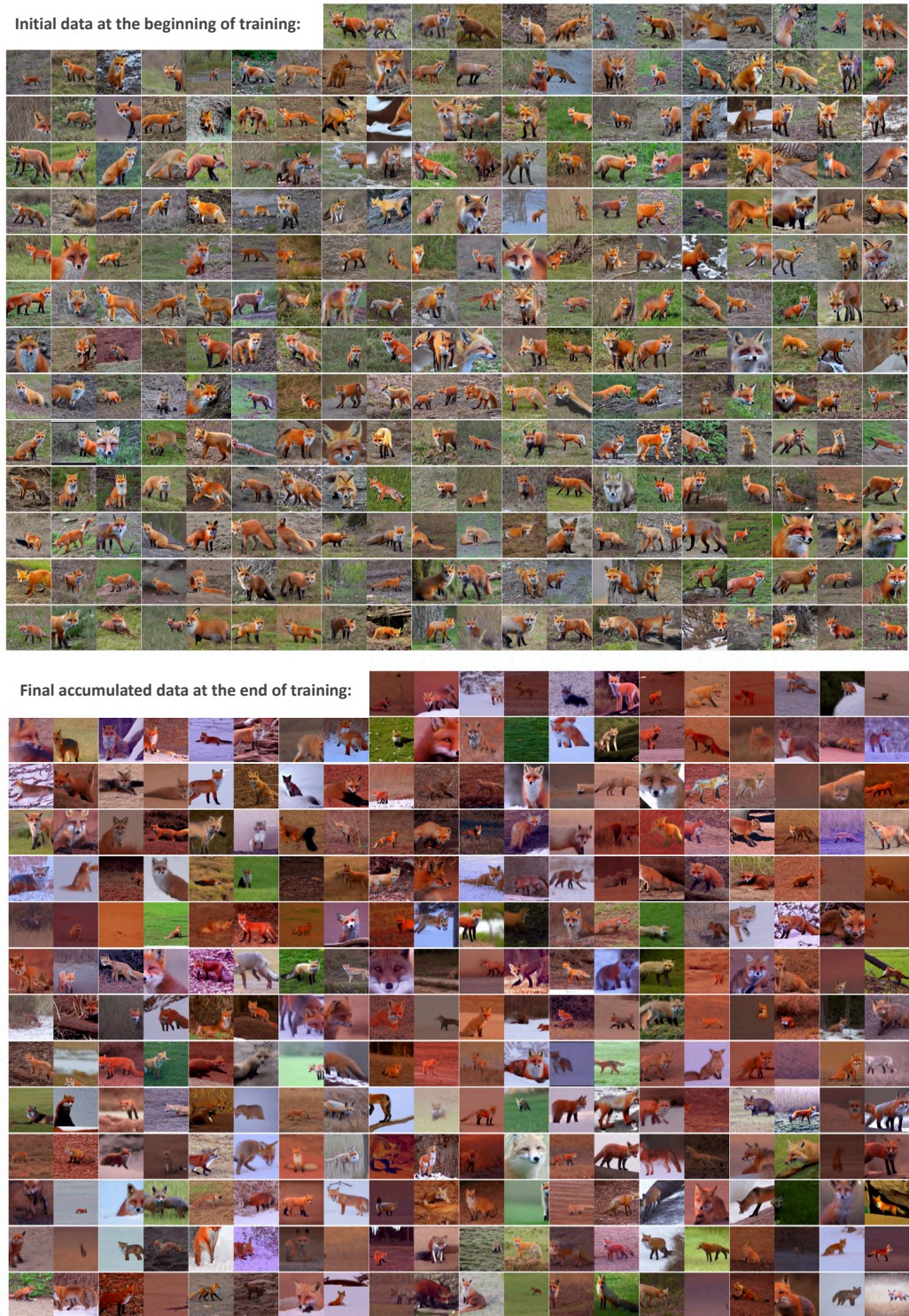

Figure 13: **Comparison of Initial and Final Training Data:** The initial training data lacks entropy guidance, as the classifier is untrained. By the end of training, the accumulated dataset contains progressively harder/diverse examples.

