# OpenReview forum: "Improving the Scaling Laws of Synthetic Data with Deliberate Practice"
_ICML.cc/2025/Conference — ICML 2025 oral_

### Official Review · Reviewer_K1M2 · 2025-03-07

**Overall Recommendation:** 3

**Summary:**

The authors propose a method for synthetically generating data based on an entropy-guided sampling of diffusion models. Their method is dynamic in that they call their entropy-guided sampling every time their model's validation accuracy plateaus. They show that their synthetically generated data is better than prior work on ImageNet-100 and ImageNet-1K. They present some theoretical motivation for their approach.

**Claims And Evidence:**

Yes the empirical claims are supported by clear evidence, but I do not see how their theory is related to diffusion models and synthetic data generation which is the main topic of their paper.

**Essential References Not Discussed:**

The theory effectively claims that selecting examples along the decision boundary is beneficial and that one should continue to update the decision boundary.  These two facts have been shown theoretically and empirically in the active learning literature in much more general settings than the ones analyzed here. See for example "Margin Based Active Learning" by Balcan et al and "Improved algorithms for efficient active learning halfspaces with massart and tsybakov noise" by Zhang et al. (for theory) and Batch Active Learning at Scale by Citovsky et al. (for empirical evidence).

**Experimental Designs Or Analyses:**

The experimental design is valid for the task they are trying to solve.

**Methods And Evaluation Criteria:**

I am confused as to why the authors use the validation set for their algorithm instead of a subset of the training set. It makes comparison to prior work more difficult as prior work reports the model performance on the test set and not the training set.

**Other Comments Or Suggestions:**

DP is not a great acronym as it is usually used for differential privacy.

**Other Strengths And Weaknesses:**

Strengths
* Leveraging the diffusion mechanism by using entropy-guided sampling presents an interesting algorithmic approach.
* Their method avoids the need to over-generate and then prune strategy that priori work has often used.
* Empirical results show positive improvement using their proposed methods.

Weaknesses
* Theory seems to be disconnected from the algorithm they present.
* Theory doesn't provide novel insights than prior theoretical work and seems to be in more restricted settings.
* Main differences with Hemmat et al 2023 are not explained in detail.
* Dynamic nature of the algorithm seems a bit ad-hoc and it would be much better if it didn't need to use an external set to decide whether to generate more synthetic data.

**Questions For Authors:**

1. What is the main difference between this work and Hemmat et al 2023?

2. The theory section seems disconnected from the setup the authors are interested in. In particular, how is the theory related to diffusion models and synthetic data generation? If I am understanding correctly, it just seems to be a linear classifier over gaussian input data?

3. I am confused by Figure 2. The black line is the strategy that selects all data, but then the x-axis is the size of the selected data?

4. For Table 2, the "IN real Val" column is the model performance on the ImageNet validation set that was used in the algorithm? If it is used in the algorithm, then we should not be comparing against it. Why didn't the authors just use a subset of the training data as their validation set so that we could more easily compare to previous work? Also why does a static method from previous work appear multiple times? Is the difference only the number of iterations run? I don't see a discussion or explanation in the main text.

5. Is there other ways to make the algorithm dynamic without having to use the train/validation set? Most dynamic selection algorithms don't make use of the validation set and are solely based on the learner so it begs the question whether this is actually needed here.

**Relation To Broader Scientific Literature:**

Synthetic data is becoming an integral tool for the broader scientific community, offering solutions for settings that are data scarce, have privacy concerns, and more generally to improve model performance.

**Theoretical Claims:**

The proofs are in the appendix and I did not carefully check them, but I do except the claims to be correct.

---

> ### Author Rebuttal · Authors · 2025-04-01
>
> We thank the reviewer for their thoughtful and detailed feedback.
>
> ---
>
> ### **On theory**
>
> **Relation to Active Learning.** Though related at a high level, our work differs from standard active learning, which focuses on querying labels for unlabeled data. We instead focus on generating useful training examples. The papers mentioned by the reviewer establish bounds which indicate that aggressive pruning restricted to a well crafted region leads to improved sample complexity for active learning. Our contributions differ: (1) We show that DP is equivalent to pruning. (2) We analyze this pruning in a toy setting using random matrix theory, deriving **exact** error curves.  (3) This analysis helps explain why DP improves sample efficiency in practice.
>
> **Connection between theory & algorithm.** The theory offers a principled explanation for DP’s success: it effectively samples from a pruned data distribution, without over-generating and discarding samples. We focus on high-dim regression as it’s analytically tractable yet captures the key aspects of the problem. While simplified, the goal of the theory is to isolate and analyze core components of the algorithm, helping explain its empirical effectiveness. We will clarify this in the revision.
>
> **It just seems to be a linear classifier over gaussian input data?** No. The theory involves linear classification over **"pruned"** Gaussian data, which significantly alters the data distribution. For example, pruning distorts the standard Marchenko-Pastur law (Lemma 4), making the analysis technically involved. This required careful use of non-standard random matrix theory tools (see Appendix). We'll clarify this in the manuscript.
>
>
>
> ### **Use of the validation set**
>
> Using a validation set for early stopping and hyperparameter tuning is standard practice [1-4]. We follow the same principle: validation is used only for model selection, not training. Like early stopping, we detect when performance saturates, but instead of stopping, we generate new data and continue. This can be seen as repeated early stopping with dynamic dataset expansion.
>
> **Fair evaluation.**
> In Table 1, we report on **both** the standard ImageNet val set (IN real Val.: commonly reported, e.g., [1, 2] since the actual test set is not public) and on the **held-out training data** (IN real tr.), which is untouched by the selection or training process in DP and serves as a true test set.
>
> **Validation use is helpful but not essential.**
> While we use the validation set to determine when to generate more data, DP does not depend on it fundamentally. Alternatives include:
>
> a. Internal signals, like training loss flattening (see Figure 5 in this anonymous link https://drive.google.com/file/d/1XbkVVsHDQyhfSJfqGkaFLC7Polb0dHZR/view).
>
> b. Predefined schedules, e.g., adding data every X epochs.
>
> c. A synthetic validation set
>
> In short, our use of validation aligns with common practice and is just one possible design.
>
> [1] Sarıyıldız et al. 2023.
> [2] Fan et al. 2024.
> [3] Dosovitskiy et al., 2021.
> [4] He et al. 2022.
>
>
> ### **Comparison with Hemmat et al.**
>
> While we build on feedback-guided generation, the two works differ significantly in their motivation, setup, and methodology. Our goal is to improve scaling laws in a **zero-shot** setting with **dynamic data generation**. In contrast, Hemmat et al. focus on class imbalance, using a **static**, one-time rebalancing step with **image-conditioned** generation.
>
> | Aspect                    | This Work                            | Hemmat et al. (2023)                  |
> |---------------------------|---------------------------------------|----------------------------------------|
> | **Primary Goal**         | Improve scaling laws of synthetic data | Handle imbalanced classification       |
> | **Use of Real Training Data**     | None           | Real examples used for conditioning    |
> | **Generation Type**      | Dynamic, repeated                     | Static, one-time                       |
> | **Diffusion Conditioning** | Class labels only                   | Image and class labels                      |
> | **Scaling Analysis**     | Theoretical + empirical               | Not studied                            |
>
>
>
> ### **Table 2 and Figure 2**
>
> In Tab 1, columns 2, 3 report baselines with varying data sizes and iterations. We include these to ensure a fair comparison with our dynamic setup, which increases data over time.
>
> In Fig 2, the x-axis shows the final data size. E.g., to train with 2^10 examples using the "top 10%" (red), we generate 2^11 and keep the hardest 10%. The "select all" (black) strategy generates and uses all 2^10.
>
> ---
> ### **DP as the acronym**
> Thanks for the suggestion. We are happy to adopt the acronym to SDP (Synthetic data with Deliberate Practice). It has a nice ring to it!  :)
>
> ---
>
> Thank you again for your constructive comments. If there's anything in particular that’s holding you back from potentially increasing your score, let us know.

---

### Official Review · Reviewer_ovd8 · 2025-03-15

**Overall Recommendation:** 4

**Summary:**

This paper focuses on the task of synthetic data generation for image classification. Specifically, traditional methods of synthetic data generation in classification tasks suffer from diminishing returns as the dataset size increases, leading to inefficient use of generated data. Inspired by the concept of deliberate practice in human learning, the authors propose a novel framework, Deliberate Practice for Synthetic Data Generation (DP). Instead of generating a static dataset upfront, DP iteratively trains a model, generates new data, and selects challenging examples to incrementally expand the training set. The authors provide theoretical justifications for their approach and conduct empirical experiments on two image classification datasets. The results demonstrate that their framework significantly improves data efficiency while reducing computational costs.

### update after rebuttal:
I thank the authors for their response and have taken into account the perspectives of the other reviewers. I will maintain my score, as it is already a positive one.

**Claims And Evidence:**

The main claim of this paper is that the proposed synthetic data generation framework improves data efficiency and reduces computational costs. I believe this claim is well-supported by evidence presented in the paper. The authors conduct experiments on two widely used datasets and show that their dynamic, multi-round data generation approach successfully produces more informative training examples. The results demonstrate that their method can achieve comparable to static methods, but with significantly fewer training samples. Additionally, comparisons with other recent synthetic data generation techniques highlight the advantages of the proposed method over traditional generate-then-filter methods in terms of data efficiency and model performance.

**Essential References Not Discussed:**

The paper discusses enough related work.

**Experimental Designs Or Analyses:**

I find the experimental setup and analysis to be well-designed and thorough. In particular, I appreciate the authors' analysis in Section 5.4, where they examine the evolution of hard examples over time to highlight how difficult samples dynamically change during training.

**Methods And Evaluation Criteria:**

I find the proposed method to be conceptually intuitive and practically applicable. The idea of iteratively refining the training data through entropy-guided sample selection is novel and well-motivated.

The evaluation criteria, including commonly used benchmarks and metrics, are reasonable for assessing the effectiveness of the proposed framework.

**Other Comments Or Suggestions:**

N/A

**Other Strengths And Weaknesses:**

Overall, I find the proposed method to be novel, and the experiments and analyses are well-conducted. One concern I have is that the authors do not provide any qualitative analysis of the generated hard samples. It would be beneficial to investigate whether the generated difficult samples are truly challenging, and whether they follow specific patterns that could offer more insights into the model’s learning process.

**Questions For Authors:**

Have you conducted any manual analysis of the generated hard samples?

**Relation To Broader Scientific Literature:**

The key contribution of this paper—multi-round training with dynamically selected hard samples—fills an important gap in synthetic data research. Many existing works focus on generating large-scale datasets and designing filtering mechanisms to remove low-quality samples. In contrast, this paper suggests that instead of generating a massive dataset and filtering it later, one can directly generate selective samples by leveraging uncertainty in the generative model.

**Theoretical Claims:**

The authors theoretically analyze in Section 4 how selecting difficult examples can improve the scaling laws of synthetic data. I have reviewed the proof and found it to be correct.

---

> ### Author Rebuttal · Authors · 2025-04-01
>
> We thank the reviewer for their detailed and encouraging review. We’re glad to hear that you found the method to be conceptually intuitive, the theoretical analysis to be sound, and the experimental setup and evaluation thorough.
>
> ### **Qualitative Analysis of Hard Samples:**
> Thank you for raising the point about qualitatively analyzing the generated hard samples. While we do include some visual examples in Figures 6 and 10, in response to your suggestion, we have conducted additional visual analysis available at this anonymous link (https://drive.google.com/file/d/1XbkVVsHDQyhfSJfqGkaFLC7Polb0dHZR/view) (can be access in incognito mode and requires no logging in):
>
> In the newly added Figures 1 and 2, we start with a pool of examples and compare the following sampling setups visually:
>
> 1. Uniform random selection from the pool
> 2. High entropy selection from the pool
> 3. Directly generating high entropy samples with DP
>
> We visually observe that:
> 1. Uniform selection does not change the distribution of the examples.
> 2. High-entropy selection results in some ambiguous or atypical examples, often featuring occlusions, rare viewpoints, complex backgrounds, or low-contrast textures; factors that increase classification uncertainty.
> 3. DP’s direct generation produces the most visually diverse and semantically rich set of samples. These include unusual lighting, texture variation, and sometimes near-failure cases; all of which challenge the model and encourage more robust learning.
>
>
> Figure 3 compares the following setups visually:
> 1. Initial samples with no entropy-guidance
> 2. DP samples with entropy-guidance earlier during training
> 3. DP samples with entropy-guidance later during training
>
> We observe that early-stage generations show mainly color diversity, while later stages exhibit a richer set of transformations, aligning with the classifier’s evolving uncertainties.
>
> Figure 4 compares the following setups visually:
> 1. Random samples from the initial data at the beginning of training for the class ‘fox’
> 2. Random samples from the final accumulated data at the end of training.
>
> The initial training data (visually) lacks diversity. By the end of training, the accumulated dataset contains progressively harder/diverse examples.
>
> ---
>
> Thank you again for your thoughtful comments and for recognizing the contributions of this work. We hope the addition of qualitative examples will further strengthen the final version. If there's anything in particular that’s holding you back from potentially increasing your score, we’d really appreciate hearing about it. We're happy to clarify or address any remaining concerns.

---

### Official Review · Reviewer_gZy1 · 2025-03-21

**Overall Recommendation:** 1

**Summary:**

This paper empirically demonstrates that "delibrate practice" is meaningful in improving the scaling law of synthetic data generation. Broadly speaking, this falls under the general umbrella of efficiently collecting as few data as possible, with the additional problem context being, a generative model is used in generating the data to be collected. Though delibrate practice was a notion related to "curiosity" and has been explored in the past, this work empirically re-examined such use in the above said context. What's encouraging is, this work presents a scaling law, not just a few case studies.

**Claims And Evidence:**

They are fine.

**Essential References Not Discussed:**

I didn't check

**Experimental Designs Or Analyses:**

They are fine.

**Methods And Evaluation Criteria:**

They are fine.

**Other Comments Or Suggestions:**

No

**Other Strengths And Weaknesses:**

I am not sure that novelty is high enough.

**Questions For Authors:**

No

**Relation To Broader Scientific Literature:**

This paper empirically demonstrates that "delibrate practice" is meaningful in improving the scaling law of synthetic data generation. Broadly speaking, this falls under the general umbrella of efficiently collecting as few data as possible, with the additional problem context being, a generative model is used in generating the data to be collected. Though delibrate practice was a notion related to "curiosity" and has been explored in the past, this work empirically re-examined such use in the above said context.

**Theoretical Claims:**

They are fine.

---

> ### Author Rebuttal · Authors · 2025-04-01
>
> Thank you for your review.
>
>
> The concept of "deliberate practice" has indeed been borrowed from psychology and has thematic connections to "curiosity". However, to the best of our knowledge, we are the first to adapt and implement this principle in the setting of training classifiers entirely on synthetic data generated from diffusion models. Rather than generating a large static dataset or pruning post hoc, we propose a dynamic approach that modifies the generative process itself to prioritize informative, high-entropy examples.
>
>
> Our work goes beyond showing isolated gains, it demonstrates that deliberate practice shifts the scaling laws of training with synthetic data, enabling significantly better performance with fewer generated examples.
>
> ---
>
> We were puzzled by the low overall rating, especially given that the review doesn’t raise specific methodological, theoretical, or experimental concerns. If there are particular issues we may have missed, we’d be happy to address them. Otherwise, if the current score was entered in error, we would be grateful if you would consider updating it.

---

> > ### Comment · Reviewer_gZy1 · 2025-04-08
> >
> > The term "deliberate practice" sounded new, but the concept is not, similar concepts had been applied for decades, for fairly unimpressive results if generalization is the goal. More recently, even in the area of LLM, similar concepts have been applied to understand cirriculum training and to arrange the order of training data, again for fairly unimpressive results judging from published materials. To this reviewer, the "novel idea" statement is an over claim.
> >
> > I agree that this work could be the first to "adapt and implement this principle in the setting of training classifiers entirely on synthetic data generated from diffusion models", which is a good thing. But while "being first" may be good for, say, CVPR, merely being first in my opinion falls short of the level of excellence required by the top 3 prestigeous AI conferences.
> >
> > The above are my reasons for low scores.
> >
> > Below is a divergence of experimental results from my expectation, but I don't have any reason to doubt the authors of doing things wrong, so below are not my reasons for low scores, they are my questions to the authors that I didn't pose in previous rounds.
> >
> > For people making LLM it was natural to expect that training more on high perplexity materials would lead to better performance. However, observation was the opposite; training too much on such data could lead to bad results. LLM practitioners still follow the rule of thumb of taking a blend of like 20% very low perplexity material, 60% intermediate perplexity material, and 20% high perplexity material, in a batch, to stablize training. Is there something like that in the experiments?

---

> > > ### Author Response · Authors · 2025-04-08
> > >
> > > Thank you for your follow-up and for taking the time to expand on your reasoning. We appreciate the opportunity to clarify the contributions of our work and address what appear to be significant misunderstandings.
> > >
> > > ### **Misrepresentation of the Contribution**
> > > We are concerned that the review mischaracterizes the novelty and core focus of our work. We want to make it explicitly clear that: We do not claim that the idea of "deliberate practice" is new. The novelty lies in how we operationalize this high-level principle for training image classifiers entirely on synthetic data, generated from a diffusion model and guided dynamically by the learner’s own uncertainty. This is not a minor reapplication of an old idea.
> > >  Our method:
> > > - Works with no access to real training image data.
> > > - Uses generation of synthetic data, not selection from a static pool.
> > > - Lets the learner update its notion of difficulty throughout training.
> > > - Leads to clear improvements over vanilla sampling in scaling behavior, as shown in our experiments.
> > >
> > > These contributions are grounded in theory and empirically validated across two datasets.
> > >
> > > ### **On LLM Parallels**
> > > In this work DP was explicitly developed for image classification. We caution against drawing direct parallels with prior LLM training strategies without careful grounding on the setup and the tasks studied. Therefore, while we appreciate the analogy, we believe that comparing prior heuristics from LLM training without rigorous validation in the vision domain could be misleading.
> > >
> > > We believe that adapting our dynamic data generation framework to LLMs could be an interesting direction for future work. But using results from prior LLM training heuristics to cast doubt on our method in a vision setting is, in our view, an invalid comparison and not a sound basis for evaluation.
> > >
> > > ### **Missing Key Parts of the Paper**
> > > We are also concerned that key parts of the paper appear to have been overlooked. In particular, the reviewer's question about the perplexity distribution in LLMs suggests that the reviewer assumes that we simply sample hard examples and train on them. That is not the case. In fact, a major focus of our analysis is to understand **how the notion of difficulty evolves as training progresses** and we have an adaptive method for this very reason. We studied the error of generated samples at different stages of training and found that: hardness is not static, it evolves throughout training and DP adapts its sampling accordingly. Also, see Figure 4 where we show that if we use a very high DP coefficient, we observe lower returns in terms of the accuracy. So very hard examples do lead to lower performance.
> > > For more details, see Sections 5.3 and 5.4 about your question in the context of learning image classifiers.
> > >
> > >  ### **Final Remarks**
> > > We respect your time, but we must express that this review does not reflect a fair or careful engagement with our work. We stand by the contribution: a simple, effective, and empirically validated method for dynamic synthetic data generation, built on well-established intuitions but delivering novel results and insights. We strongly encourage a more careful reading of the paper and a reconsideration of the score in light of the clarifications above.

---

### Official Review · Reviewer_vJVZ · 2025-03-21

**Overall Recommendation:** 4

**Summary:**

The authors introduce a new methodology called "Deliberate Practice for Synthetic Data Generation" (abbreviated DP) to train a machine learning model for classification using entirely synthetic samples generated from a pre-trained diffusion model. Instead of generating many samples and pruning, the method uses an additive correction to the score to prioritize samples which are most challenging for the model; this correction is updated dynamically during training and is based on the entropy of the model on the sample at the time where the data is generated. The authors justify their approach of selecting difficult examples via theory for high-dimensional linear regression. Then, they perform some experiments: (1) comparing DP to another ("static") approach of pre-generating all data and training on a fixed synthetic dataset; (2) comparing DP to previous methods [1, 2]; (3) comparing DP to another ("pruning") approach of pre-generating data and then pruning via model entropy; (4) answering the scientific question of whether models trained with DP change their (entropy-based) difficulty estimates over the training trajectory. For experiments (1)-(3) the DP method generally performs uniformly better than the chosen alternatives; for experiment (4) it confirms that models trained with DP indeed change the difficulty estimates.

[1] Sarıyıldız, M. B., Alahari, K., Larlus, D., and Kalantidis, Y. Fake it till you make it: Learning transferable representations from synthetic imagenet clones. In CVPR, 2023.

[2] Fan, L., Chen, K., Krishnan, D., Katabi, D., Isola, P., and Tian, Y. Scaling laws of synthetic images for model
training... for now. In CVPR, 2024.

## Update After Rebuttal

The rebuttal clarified the theoretical/conceptual points I had raised. Because of that, I think the work is a good contribution and should be accepted. The other point I raised (which seems to be echoed by the other reviewers) is that the experimental section's scope is a little lacking; I had expected more datasets/diffusion models in order to verify the scientific claims, while other reviewers comment on specific evaluations and comparisons to other work. Still, the experimental results seem overall reasonable to me. For that reason I recommend acceptance.

**Claims And Evidence:**

There are several claims being made in the paper:

(1) the asymptotics of the test error of the toy linear regression setup in Section 4;

(2) the advantage of DP (in the classification setting) over more trivial methods for synthetic data generation like "pruning" and "static" approach;

(3) the advantage of DP (in the classification setting) over previously proposed methods for synthetic data generation;

(4) the claim that the same sample may be considered more or less difficult by a model over the DP training.

Of these, (1) is a mathematical claim and it seems adequately backed up via proofs in the Appendix A. (2) and (3) are empirical claims about method performance, and hold on the provided testbed (trained on varying number of samples generated from the same diffusion model LDM-1.5, and evaluated on subsets of ImageNet 1K). Still, in order to back up the more general claims it may help to generate from different diffusion models and evaluate on different datasets. (4) is a scientific claim and straightforwardly backed up.

**Essential References Not Discussed:**

I don't know any significant omitted references.

**Experimental Designs Or Analyses:**

The experimental designs and mechanisms (including things like hyperparameters) seem well-described to me; while many details (and ablations, etc.), are not in the main body, they are in Appendix B. The analysis of the above experiments in Section 5 and Appendix B also seems sound.

**Methods And Evaluation Criteria:**

The methods are: training using DP (or other methods) on samples generated from a single diffusion model LDM-1.5 (except for a small ablation in the appendix) and evaluated on different subsets of ImageNet-1K. These methods and evaluations make sense and are standard, though (as remarked above) to truly back up the claims, it might help to use more evidence using different generative models (especially since LDM-1.5 was picked specifically because DP performs best with it relative to the other tested generative models) and/or different evaluation testbeds.

**Other Comments Or Suggestions:**

Some nitpicks:
- In equation (5) I think there should be a $(t)$ superscript for all model uses.
- Section 3 title: "Deliberate" should be capitalized.
- The paragraph titles have inconsistent formatting; some are capitalized ("Asymptotic Behavior of the Test Error") while most aren't.
- Some Typos in Appendix B, such as the section title "Additional Experimental".

**Other Strengths And Weaknesses:**

- The main strengths are the improvement of DP over previous methods, as well as its simplicity.
- The main weaknesses are the lack of diversity in experimental setups (discussed previously), and the lack of connection between the theory and practice, which operate in two different regimes (classification vs regression, proportional scaling vs not, etc) and so it is unclear what messages or guidance to take from the theoretical part.

**Questions For Authors:**

No questions at this time.

**Relation To Broader Scientific Literature:**

This paper broadly intersects with two different areas:
- Synthetic data and its use to train neural networks (and associated with challenges such as model collapse, bias of the generator, etc).
- Conditional diffusion models (for generating the synthetic data in this work).
- Active learning and continual learning (which also involve tailoring data to improve learning trajectory).
- High-dimensional statistics (corresponding to the theory component.)

**Theoretical Claims:**

While I did not carefully check the proofs extremely carefully, they seem to be valid (and there is experimental evidence on toy datasets to confirm their correctness). A minor issue is that in Theorem A.1 it's not specified in which order the limits $\lambda \to 0$ and $n, d \to \infty$ with $d/n \to \phi$ are taken.

Some slightly larger problems with the theoretical component of this work are:
1) The rate in Theorem 1 the main body is not very "clean"; when I look at the rate I have no idea what order of magnitude I expect for the test error. It could be possible to work it out using asymptotic expansions of $\Phi$ and the other terms, but it would be very helpful to understand the rate approximately in the main body (while keeping the more complicated part for the appendix, of course), and it could help make some qualitative insights very straightforward.
2) The use of the proportional scaling regime for linear regression is slightly un-natural (to me); in the experiments you're only scaling data, not dimension, and so it might make sense to consider fixed $d$ and large $n$. What kinds of asymptotics are achieved there? Are they interesting? It would be useful to comment.
3) More generally, the linear regression problem in Section 4 is a toy problem, and the practice uses cross-entropy losses and measures the difficulty totally differently from the sign-of-inner-product way that's given in Section 4. (Of course, the real data isn't generated by a Gaussian and linear classifier either.) While I appreciate the considerable difficulty of expanding the analysis to this more practical setting, these differences should be remarked.

These may cause the (considerable) difference between real and predicted accuracy in Figure 4 (c) --- it would be helpful to comment on that and bring the theory a little closer to practice.

---

> ### Author Rebuttal · Authors · 2025-04-01
>
> We thank the reviewer for the detailed feedback. We're glad that you found the method conceptually clear. Also, thank you for the helpful proofreading suggestions. We will incorporate these corrections in the final version.
>
> ---
>
> **Connection Between Theory and Practice:** We agree that the linear regression setup in Section 4 is a simplified version of the full setup used in practice. The goal of this section is not to precisely model the real-world setting, but to isolate and analyze the principle of prioritizing difficult examples based on entropy or uncertainty. We will include a short discussion highlighting the limitations and interpretability benefits of the theoretical analysis in the simple setup. Also, note that even though loss used in training the linear model is regression loss (squared L2), the learned model is used as a classifier and we analyze classification accuracy (see Section 4). The use of linear classifiers fitted with squared loss is common in ML theory (Couillet & Liao, 2022; Liao & Mahoney, 2021; Firdoussi et al., 2024).
>
> **Unpacking Theorem 1.** The point of the theorem is to showcase that the test error can be written analytically as a function all problem parameters: parametrization rate $\phi$, regularization parameter $\lambda$, pruning ratio $p \in (0,1)$, cosine of angle between the pruning direction and the ground-truth labelling function $\rho$, etc. Analytic expression provided is defined via concrete functions like $\Phi$, and the spectral functions (linked to Marchenko-Pastur Law) introduced earlier and can be used to numerically compute / simulate the shape of the theoretical error curve in different regimes corresponding to different settings of the problem parameters. We agree with the reviewer that expanding the Phi function can provide some quantitative insights, at least in setting regimes of its argument, especially: large +ve, large -ve, and small values.
>
> Finally, note that in the unregularized (ridgeless) case $\lambda \to 0^+$, the aforementioned analytic expression simplifies drastically, as shown in Corollary 1 (Appendix).
>
> In Appendix A.1, the order of the limits is: first let $d,n \to \infty$ such that $d/n \to \phi$. Then, let $\lambda \to 0^+$.
>
> **Proportional scaling in linear regression.** This is actually standard in high-dimensional statistics and learning theory. Here, such a scaling allows us to capture important regimes: interpolation threshold (corresponding to $\phi=p$), under-parametrized (phi < p), over-parametrized ($\phi > p$), extreme under-parametrized ($phi \to 0^+$), extreme over-parametrized ($\phi \to \infty$) regimes while keeping the analysis tractable (via the tools of random matrix theory).
>
> The reviewer’s statement that we are “only scaling data” is inaccurate. Observe that $d,n \to \infty$ such that $d/n\to \phi$ can be equivalently written as $n = \phi d$ (rounded to an integer), $d \to \infty$. Therefore, we can fix $d$ to a large value and study the effect of $\phi$, as done in the experiments. Other large values of $d$ would yield a similar experimental picture. Everything about scale is completely captured in the $\phi$ parameter. We shall make this clearer in the manuscript.
>
>
> ---
>
> ### **Experimental Diversity:**
> We appreciate your suggestion to evaluate DP across a broader range of generative models and datasets. While we included comparisons in Appendix B.1 using other diffusion models, and have already reported results on two datasets (ImageNet-100 and ImageNet-1K) we acknowledge that this could have been more clearly emphasized in the main paper. We will revise the manuscript to better highlight the diversity already present in our evaluation.
>
>
> We also want to clarify that LDM-1.5 was **not chosen because DP performs best with it**. Rather, we selected it because it consistently provides the strongest baseline performance in static data generation setups, independent of DP (as shown in Appendix B.1). This makes it a compelling and fair testbed. This model choice is in-line with prior work (e.g., Astolfi et al., 2024 and  Fan et al., 2024), which uses LDM-1.5 as it produces more diverse samples than alternatives which is a desirable property when training classifiers on synthetic data.
>
>
> More broadly, we do not expect DP to be limited to a particular diffusion model or sampling strategy. Our approach is a lightweight modification of the sampling process, similar in spirit to classifier guidance, that leverages the model's entropy to influence which samples to be generated. As long as the sampler provides an approximation of the denoised sample x_0, DP can be applied by using the downstream classifier to adjust the sampling trajectory. We will clarify this generality in the revised version.
>
> ---
>
> Thank you again for your constructive comments. If there's anything in particular that’s holding you back from potentially increasing your score, let us know.

---

> > ### Comment · Reviewer_vJVZ · 2025-04-01
> >
> > Thanks for clarifying the following.
> >
> > > The purpose of the theory
> >
> > This point now makes sense to me. There is learning theory work with the cross-entropy loss (e.g. [1]), but I acknowledge it (maybe significantly) increases the difficulty of the analysis. While it would have been great if the work were actually extended to this setting, even the current simplified setting seems to capture good insights.
> >
> > > The proportional scaling regime
> >
> > This point also makes sense to me, thank you for clarifying. You may want to put this explanation in the camera-ready version, as I found it a very nice and clean justification.
> >
> > > Other theoretical details
> >
> > Thanks for the various clarifications on the order of the limits and the asymptotic order of the test error; I would really encourage these to be put in the camera-ready version of the paper.
> >
> > > The experimental diversity and choice of data/samplers for synthetic data
> >
> > Here I know that the previous work uses LDM-1.5 (and thank you for clarifying why it is picked in this paper, I originally thought that Appendix B.1 used Deliberate Practice to compare the models). However, the claim made in the paper and in the rebuttal is that Deliberate Practice generalizes to different classification datasets (e.g., in the paper ImageNet-100 and ImageNet-1K) and diffusion models. In principle this is obviously true --- the proposed method does not specify any required number of classes nor requires a specific diffusion model, so there is no concrete obstruction to applying Deliberate Practice using any model and classification dataset. But to demonstrate the claimed generalization in practice, I would have expected a greater diversity of evaluations across different datasets and models.
> >
> > Note that the evaluated data is all subsets of ImageNet-1K and evaluated models are all within the LDM model family. Also, the comparison between diffusion models in Appendix B.1 aren't about Deliberate Practice but on another proxy task which seems to measure the diversity/prompt adherence of the samples. So it does not seem to me to be really sufficient to demonstrate the efficacy of Deliberate Practice using different diffusion models.
> >
> > As a result of the latter point, I keep my score.
> >
> > [1] Wu, Jingfeng, Vladimir Braverman, and Jason D. Lee. "Implicit bias of gradient descent for logistic regression at the edge of stability." Advances in Neural Information Processing Systems 36 (2023): 74229-74256.

---

> > > ### Author Response · Authors · 2025-04-08
> > >
> > > Thank you for your reply! We appreciate your positive assessment and your thoughtful engagement with both the paper and our clarifications. We're glad the theoretical points now make sense, and we will be sure to include those clarifications in the camera-ready version.
> > >
> > > We’d like to add one final clarification regarding the results in Appendix B.1. The comparison there does not measure the diversity or prompt adherence of the generated samples. Instead, it reports the validation accuracy of four different classification models, each trained on synthetic data generated by a specific LDM variant. This experiment serves to validate that our choice of generative model (LDM-1.5) is fair and appropriate for the task of image classification.
> > > As mentioned earlier, we do not expect Deliberate Practice to fail on any of the generative models evaluated. Since DP only modifies the DDIM sampling process, and that process is applicable across all of these diffusion models, the method is fully compatible and should retain its benefits.
> > >
> > > While we acknowledge that additional datasets and model families would further strengthen the generalization claims, we hope this clarifies the rationale behind our current setup and supports the fairness of the experimental evaluation.

---

### Official Review · Reviewer_rVX6 · 2025-03-21

**Overall Recommendation:** 5

**Summary:**

This work addresses the challenge of improving data size scaling laws for models trained on synthetic data. In particular, it uses the intuitive idea of generating more synthetic data where the model being trained (referred to as the learner) has high entropy. To do so, the paper relies on a setting where the synthetic data is generated using a 'Denoising Diffusion Implicit Models' & modifying the generation to prefer samples where the learner has high entropy. This approach is illustrated to work in a toy theoretical setting as well as through real-world experiments on IN-100 & IN-1K.

**Claims And Evidence:**

Yes.

1.  The claim of the proposed sampling scheme modifying the sampling distribution is supported by the steps illustrated in equations 3-6.

2. The claim of this approach improving scaling laws is evidenced by the experiments on ImageNet.

3. The toy theoretical example serves to further illustrate this point.

**Essential References Not Discussed:**

I believe there should be an inclusion of references to earlier works on data pruning including, but not limited to:

- Coleman, Cody, et al. "Selection via proxy: Efficient data selection for deep learning." arXiv preprint arXiv:1906.11829 (2019).
- Mindermann, Sören, et al. "Prioritized training on points that are learnable, worth learning, and not yet learnt." International Conference on Machine Learning. PMLR, 2022.
- Pooladzandi, Omead, David Davini, and Baharan Mirzasoleiman. "Adaptive second order coresets for data-efficient machine learning." International Conference on Machine Learning. PMLR, 2022.
- Yang, Yu, Hao Kang, and Baharan Mirzasoleiman. "Towards sustainable learning: Coresets for data-efficient deep learning." International Conference on Machine Learning. PMLR, 2023.
- Joshi, Siddharth, et al. "Data-efficient contrastive language-image pretraining: Prioritizing data quality over quantity." International Conference on Artificial Intelligence and Statistics. PMLR, 2024.

**Experimental Designs Or Analyses:**

Yes, this is sound.

**Methods And Evaluation Criteria:**

Yes, I believe the datasets & baselines used for this method are fair & comparing to SOTA.

**Other Comments Or Suggestions:**

I think the paper could benefit from a more detailed background section on DDIM. Personally, since I was not familiar with this, I had to refer prior work to understand this, before I could fully understand the paper & it would be better if the paper could be read in a self-contained manner.

**Other Strengths And Weaknesses:**

N/A

**Questions For Authors:**

Have the authors considered a similar idea for training language models? Do they have any initial thoughts on how such an approach would be extended to that setting?

**Relation To Broader Scientific Literature:**

I think this is a useful practical tool in the training of image models using synthetic data. The idea & implementation of modifying a diffusion model to generate examples in high-entropy regions for a classifier being trained is extremely intuitive, simple & effective.

**Theoretical Claims:**

Yes, I checked the theorem in the main paper and this looks correct.

---

> ### Author Rebuttal · Authors · 2025-04-01
>
> Thank you for the thoughtful and encouraging review, we are very pleased that you found the paper intuitive, practical, and effective.
>
>
> ### **Related Work and Data Pruning:**
> Thanks for pointing out these relevant papers. While our method focuses on improving training via generation of informative synthetic data rather than selection or pruning of real data, both lines of work share the goal of increasing data efficiency by concentrating training on examples that contribute most to learning. In particular, works like Coleman et al. (2019) and Mindermann et al. (2022) develop data selection strategies based on proxies for example utility, such as gradient norms or learnability. Similarly, Pooladzandi et al. (2022) and Yang et al. (2023) propose coreset-based methods that aim to retain the most informative or representative points. Joshi et al. (2024) further emphasize the importance of data quality over quantity, which aligns closely with our motivation, though we operationalize this through generative modeling and entropy feedback rather than dataset filtering as discussed in Sec 5.3. We view our approach as complementary: instead of selecting from a fixed pool of data, we dynamically generate examples in regions where the model exhibits high uncertainty, effectively synthesizing the types of data these methods might prioritize. We will include this discussion in the revised related work section.
>
>
> ### **DDIM Background:**
> We agree that the paper would benefit from a more complete and self-contained explanation of DDIM. Due to space constraints, we had to keep the background section concise, but we will expand it in the final version. We also want to clarify that while our experiments use DDIM, the proposed method is not limited to this specific sampler. Our approach relies on accessing an approximation of the denoised sample x_0 at each step, which is also available in other diffusion samplers such as DDPM and its variants. As long as the sampler provides a usable x_0 ​ estimate, the entropy-guided feedback mechanism remains cheap and can be applied efficiently. We will emphasize this more clearly in the revised version.
>
>
> ### **Language Model Extension:**
> This is a very relevant direction which we have been pondering about. While our current work focuses on image generative models, the principle of DP can be extended to language. One key reason this approach works efficiently with diffusion models is that we have access to an approximation of the clean sample x_0  at each step, which enables us **to steer the sampling process before the generation ends**. This intermediate visibility during generation allows us to influence the generation process. Extending this idea to autoregressive language models is less straightforward since generation typically proceeds token-by-token with limited ability to revise earlier outputs. However, the emergence of language diffusion models opens up a promising path [1]. These models offer a latent trajectory similar to image diffusion models and may allow for entropy-guided sampling. We see this as a compelling direction for future work and are actively thinking about how DP could be incorporated into language diffusion models for synthetic data training.
>
>
> [1] Nie, Shen, et al. "Large Language Diffusion Models." arXiv preprint arXiv:2502.09992 (2025).
>
> We are very open to further discussion and would be happy to address any other questions.

---

> > ### Comment · Reviewer_rVX6 · 2025-04-02
> >
> > I continue to strongly recommend this paper for acceptance.

---

> > > ### Author Response · Authors · 2025-04-08
> > >
> > > Thank you for your continued support and for the thoughtful feedback throughout the review process. We appreciate your recommendation and are pleased that you found the work valuable.

---

### Decision · Program_Chairs · 2025-05-01

**Decision:**

Accept (oral)

**Comment:**

This paper applies the principle of Deliberate Practice to effectively generate synthetic data with Denoising Diffusion Implicit Models for image classification tasks. There are five reviews received for the paper with a distribution of (1, 3, 4, 4, 5). The authors have the opportunity to engage with reviewers during the rebuttal phase, and clarify on several fronts, especially the contributions and experimental settings. The only negative review was abstract at the very beginning, and even after one round of rebuttal, the review can still improve on clarity and be more constructive itself. While I appreciate reviewer's efforts in reading the paper and discussing with the authors, I would have put more weights on the negative review if a few clear references are given on the main argument of limited novelty. As ICML this year particularly emphasize correctness over novelty, this paper is a clear acceptance.

During the discussion, while reviewers acknowledge the paper is generally well written, it appears the presentation can be improved for clarifying contributions and experimental settings. In fact, there are some remaining concerns on the fairness of experiments, which has not been clarified with the limited number of communications during rebuttal. The paper can be further strengthened by considering generative tasks beyond classification, language and multi-modal tasks, while most of the reviewers acknowledge the current contribution and happily accept the paper.